# Artificial photosynthetic cells with biotic–abiotic hybrid energy modules for customized CO$_2$ conversion

Feng Gao[1,4], Guangyu Liu[1,4], Aobo Chen[1], Yangguang Hu[1], Huihui Wang[1], Jiangyuan Pan[1], Jinglei Feng[1], Hongwei Zhang[1], Yujie Wang[1], Yuanzeng Min [1], Chao Gao [1] ✉ & Yujie Xiong [1,2,3] ✉

Programmable artificial photosynthetic cell is the ultimate goal for mimicking natural photosynthesis, offering tunable product selectivity via reductase selection toward device integration. However, this concept is limited by the capacity of regenerating the multiple cofactors that hold the key to various reductases. Here, we report the design of artificial photosynthetic cells using biotic–abiotic thylakoid–CdTe as hybrid energy modules. The rational integration of thylakoid with CdTe quantum dots substantially enhances the regeneration of bioactive NADPH, NADH and ATP cofactors without external supplements by promoting proton-coupled electron transfer. Particularly, this approach turns thylakoid highly active for NADH regeneration, providing a more versatile platform for programming artificial photosynthetic cells. Such artificial photosynthetic cells can be programmed by coupling with diverse reductases, such as formate dehydrogenase and remodeled nitrogenase for highly selective production of formate or methane, respectively. This work opens an avenue for customizing artificial photosynthetic cells toward multifarious demands for CO$_2$ conversion.

Natural photosynthetic organisms integrate solar energy-converting and biocatalytic modules within organelles, such as chloroplast, to convert atmospheric CO$_2$ and water into carbohydrates by sunlight, forming the essential basis for life's existence on earth[1]. Imitating nature to construct programmed artificial photosynthetic cells depicts a beautiful blueprint for achieving CO$_2$ conversion that meets human demands[2–4]. In contrast to the fact that the natural photosynthesis of a living organism is responsible for growth and reproduction, artificial photosynthetic cells can be rationally designed to operate more efficiently in converting CO$_2$ into valuable fuels and chemicals. Albeit photosynthetic single-celled organisms, such as cyanobacteria, have been developed for CO$_2$ conversion, commonly the carbonaceous

product over one organism is nonoptional and cannot be customized[5,6]. Significantly, the goal of artificial photosynthetic cells for CO$_2$ conversion is to achieve customizable carbonaceous products toward multifarious demands. Beyond that, artificial photosynthetic cells hold the key to finally simulating the organizational morphology and features of natural photosynthetic organisms, opening an avenue for constructing devices toward practical applications.

Inspired by natural photosynthetic organisms, an artificial photosynthetic cell should consist of three components: energy module, biocatalytic module and cofactor[1]. A typical photosynthetic process starts with the light-harvesting by an energy module to generate electrons for energy-rich chemical cofactors, which subsequently

---

[1]Hefei National Research Center for Physical Sciences at the Microscale, Collaborative Innovative Center of Chemistry for Energy Materials (iChEM), School of Chemistry and Materials Science, University of Science and Technology of China, 230026 Hefei, Anhui, China. [2]Institute of Energy, Hefei Comprehensive National Science Center, 350 Shushanhu Rd., 230031 Hefei, Anhui, China. [3]Anhui Engineering Research Center of Carbon Neutrality, College of Chemistry and Materials Science, Key Laboratory of Functional Molecular Solids, Ministry of Education, Anhui Normal University, 241002 Wuhu, Anhui, China. [4]These authors contributed equally: Feng Gao, Guangyu Liu. ✉e-mail: gaoc@ustc.edu.cn; yjxiong@ustc.edu.cn

power the biocatalytic module to perform catalytic reactions. The key to tailoring $CO_2$ conversion toward the desired carbonaceous product is the biocatalytic module, namely the enzymes, within the artificial photosynthetic cell. Several $CO_2$ convertases, such as formate dehydrogenase, have been employed in photoenzymatic systems for catalytic $CO_2$ conversion[7]. To display their capacities for biocatalytic $CO_2$ conversion, different $CO_2$ convertases require distinct energy-rich chemical cofactors, typically reduced nicotinamide adenine dinucleotide phosphate (NADPH), reduced nicotinamide adenine dinucleotide (NADH) and adenosine triphosphate (ATP)[8]. As such, customization of products in $CO_2$ conversion can be programmed by loading diverse reductases into artificial photosynthetic cells. It should be noted that, when the cofactors power the biocatalytic reductases, they will be oxidized and lose their activity for long-term operation[9]. To this end, it requires the regeneration of multiple cofactors in artificial photosynthetic cells. However, developing one photoenzymatic platform that can achieve efficient regeneration of multiple cofactors remains a giant challenge.

The regeneration of cofactors is governed by energy-converting modules. Currently, two categories of energy modules—abiotic and biotic ones have been developed for this purpose. The abiotic energy modules take advantage of man-made materials (e.g., semiconductors) to harvest light outpacing natural photosynthesis[10]. Nevertheless, most likely due to the deficient biocompatibility of synthetic materials, the electron transfer dynamics between light-harvesting energy modules and biological cofactors are severely hindered due to the lack of electronic coupling. More seriously, the bioactive cofactors and enzymes tend to suffer from threats regarding deactivation. Consequently, the abiotic energy modules commonly power NADH-dependent reactions (Supplementary Table 1), while lacking the capacity of regenerating multiple cofactors (e.g., NADPH and ATP)[11]. On the other hand, as a typical biotic energy module in natural photosynthesis, thylakoid (Tk) offers unparalleled opportunities as a promising energy module for cofactors regeneration considering its excellent mechanical stability, cost-effective extraction, and more importantly, distinct capacity for co-regenerating NADPH and ATP by sunlight[2,12]. Albeit they have shown great potential, such thylakoid-based biotic energy modules generally suffer from low utilization of light[7], which not only limits the efficiency of the whole photosynthetic process but also poses a challenge for increasing the supply of photogenerated electrons. Particularly, although the supply of external electron-transport proteins can increase efficiency, it brings higher costs and still remains inefficient in regenerating NADH[2,12].

To address the challenges above, we thus think about the possibility of whether NADH could be regenerated by increasing the supply of photogenerated electrons while promoting the regeneration of NADPH and ATP. In this regard, we need to develop a specific energy module to increase the supply of photogenerated electrons, which eventually breaks the existing bottlenecks. Increasing the supply of photogenerated electrons can be achieved by rationally combining thylakoid with inorganic light-harvesting nanomaterials[13]. As a class of materials for harvesting visible light, quantum dots (QDs) have the benefits that their light absorption and electronic band structure can be more readily tuned (Supplementary Fig. 1). Moreover, QDs have more negative conduction band potential compared to the corresponding nanoparticles, which could facilitate the transfer of photogenerated electrons to thylakoid[14–16]. Therefore, we aim to verify whether the rational integration of thylakoid with QDs could substantially enhance the supply of photogenerated electrons for efficient regeneration of multiple cofactors. Once such a possibility is proven, the realization of programmable artificial photosynthetic cells via microfluidics will become feasible.

In this work, as a proof of concept, we designed a programmable artificial photosynthetic cell by interfacing thylakoid with model CdTe QDs as a special biotic–abiotic hybrid energy module, in which the energy module offered an efficient capability for multiple cofactors regeneration to power diverse photoenzymatic $CO_2$ conversion (Fig. 1). The prepared hybrid energy module (namely, Tk−CdTe) enabled efficiently regenerating multiple bioactive cofactors (NADH, NADPH and ATP) without requiring external supplements (i.e., complementary protein and electronic mediators). Specifically, the implementation of CdTe QDs on thylakoid significantly enhanced the regeneration of bioactive NADPH, NADH and ATP cofactors by supplying photogenerated electrons and promoting proton-coupled electron transfer. As a result, three NADPH-, NADH- and ATP-dependent $CO_2$ reductases were adopted as models to demonstrate multifarious $CO_2$ conversion in our designed artificial photosynthetic cells, with the customizable products of formate and methane, respectively. This work provides a promising approach to engineering artificial photosynthetic cells by employing the biotic–abiotic hybrid energy modules to drive multiple cofactors regeneration, which sets up a platform for photoenzymatic $CO_2$ conversion toward customizable carbonaceous products. With this approach, we expect that more $CO_2$ reductases can be integrated into artificial photosynthetic cells in the future, achieving a variety of selective products for $CO_2$ conversion.

## Results
### Characterizations of thylakoid−CdTe energy modules
To achieve programmable artificial photosynthetic cells, the multiple cofactors-dependent photoenzymatic $CO_2$ conversion should be accomplished. To this end, we designed a platform for regenerating multiple cofactors by integrating thylakoid (Tk) with CdTe quantum dots (QDs) as hybrid energy modules. CdTe QDs were prepared in aqueous solution by using mercaptopropionic acid as a capping ligand[17], showing a good aqueous dispersibility with the average size of $3.86 \pm 0.8$ nm (Supplementary Fig. 2). The obtained CdTe QDs are negatively charged due to the synthetic method, as indicated by the zeta potential of $-50.1 \pm 0.56$ mV via dynamic light scattering (DLS) measurements. To facilitate the combination of thylakoid with CdTe, the CdTe QDs were functionalized with a cationic polymer poly(diallyldimethylammonium chloride) (PDADMAC) to acquire positive surface charge with the zeta potential of $30.1 \pm 4.89$ mV (namely, CdTe$^+$)[18]. Transmission electron microscopy (TEM) images reveal that the size, morphology and dispersibility of CdTe$^+$ are well maintained (Supplementary Figs. 2 and 3).

The thylakoid was isolated from fresh spinach by a Percoll and osmotic shock method[2]. Upon the isolation of thylakoid, the zeta potential and location of plasma membrane were identified by cell membrane staining, which confirms that thylakoid has been completely isolated from the membrane of chloroplast (Supplementary Fig. 4). Blue native−polyacrylamide gel electrophoresis (BN-PAGE) assay and UV−Vis absorption spectroscopy were further performed to verify their structural integrity and light-harvesting capability (Supplementary Fig. 5)[12]. These results manifest that the structure and light-harvesting function of the obtained thylakoid are well preserved. Notably, the presence of negative charges on the thylakoid enables its interaction with CdTe$^+$ through electrostatic force. As such, we can modify thylakoid with CdTe$^+$ at different mass ratios (namely, Tk−CdTe 2:1, Tk−CdTe 1:1 and Tk−CdTe 1:2). The Tk−CdTe possesses the positively charged surface, obviously different from pristine thylakoid, which is ascribed to the surface modification with CdTe$^+$ (Supplementary Fig. 6). By increasing the adding amount of CdTe QDs, the loading efficacy of CdTe$^+$ in Tk−CdTe decreases as determined by inductively coupled plasma atomic emission spectrometry (ICP-AES) (Supplementary Fig. 7, Supplementary Table 2), suggesting the loading limit of CdTe$^+$ on thylakoid through electrostatic interaction.

The loading of CdTe QDs on thylakoid was further confirmed by scanning electron microscopy (SEM) and TEM. Upon forming the hybrid structure of Tk−CdTe, the surface and margin of sample

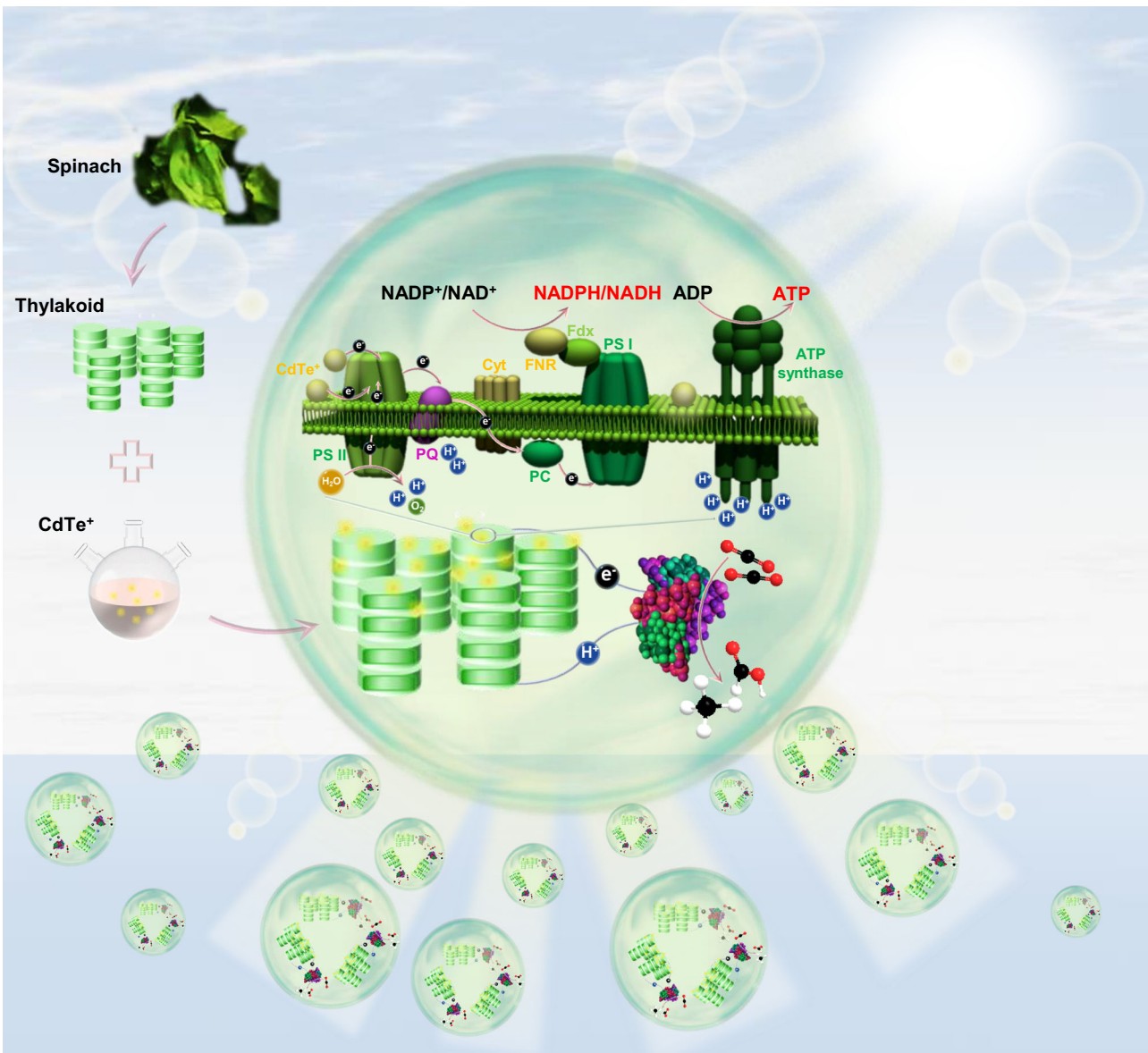

**Fig. 1 | Schematic illustration of programmable artificial photosynthetic cell.** The cell is designed on the biotic–abiotic energy modules of thylakoid–CdTe. Upon light illumination, the energy modules regenerate bioactive NADPH, NADH and ATP cofactors, which can be coupled with various cofactors-dependent CO₂ reductases to drive CO₂ conversion. The energy modules, multiple cofactors and reductases are encapsulated within cell-size microdroplets to form artificial photosynthetic cells, which can be energized by light to power enzymes for CO₂ conversion.

became rough as compared with pristine thylakoid (Fig. 2a). Energy-dispersive X-ray spectroscopy (EDS) mapping revealed that the sample contains the elements of magnesium from thylakoid as well as tellurium and cadmium of CdTe QDs (Fig. 2b). Moreover, we recognized that the formation of Tk−CdTe hybrid structure through electrostatic interaction ensured homogeneous deposition of CdTe QDs on thylakoid, and negligible large CdTe⁺ aggregates were observed (Supplementary Figs. 3 and 8). The adsorption sites for CdTe⁺ on thylakoid membrane are likely the negatively charged carboxyl in lipids or photosynthetic protein. Such homogeneous distribution of CdTe QDs on thylakoid will be beneficial for light harvesting. From the optical spectra (Supplementary Fig. 9), Tk−CdTe possesses a broad range of light absorption that well combines those of thylakoid and CdTe QDs.

It is worth noting that, with the increase of CdTe⁺ concentration, the final morphology of Tk−CdTe did not exhibit significant distinction (Fig. 2a, b). We further employed confocal laser scanning microscopy (CLSM) to visualize distribution patterns. As displayed in Fig. 2c, the

surface of Tk−CdTe remained brightly yellow fluorescent substances despite the absence of any staining process, indicative of colonization of CdTe QDs on thylakoid. Looking into the enlarged fluorescent region, we recognized that yellow fluorescent CdTe QDs are located around the perimeter of red fluorescent thylakoid (Supplementary Fig. 10). Taken together, these results reveal that CdTe QDs have been successfully uniformly coated on the surface of thylakoid, forming the designed Tk−CdTe energy modules.

## Light-driven regeneration of multiple cofactors
Upon the successful construction of Tk−CdTe energy modules, we sought to explore the feasibility of implementing energy modules in the regeneration of multiple cofactors. To examine the light-driven regeneration of NADPH, NADH and ATP cofactors, we followed the well-established methods for these cofactor molecules in literature (Supplementary Fig. 11). The production of NAD(P)H can be conveniently monitored by observing the absorbance at 340 nm[19], while

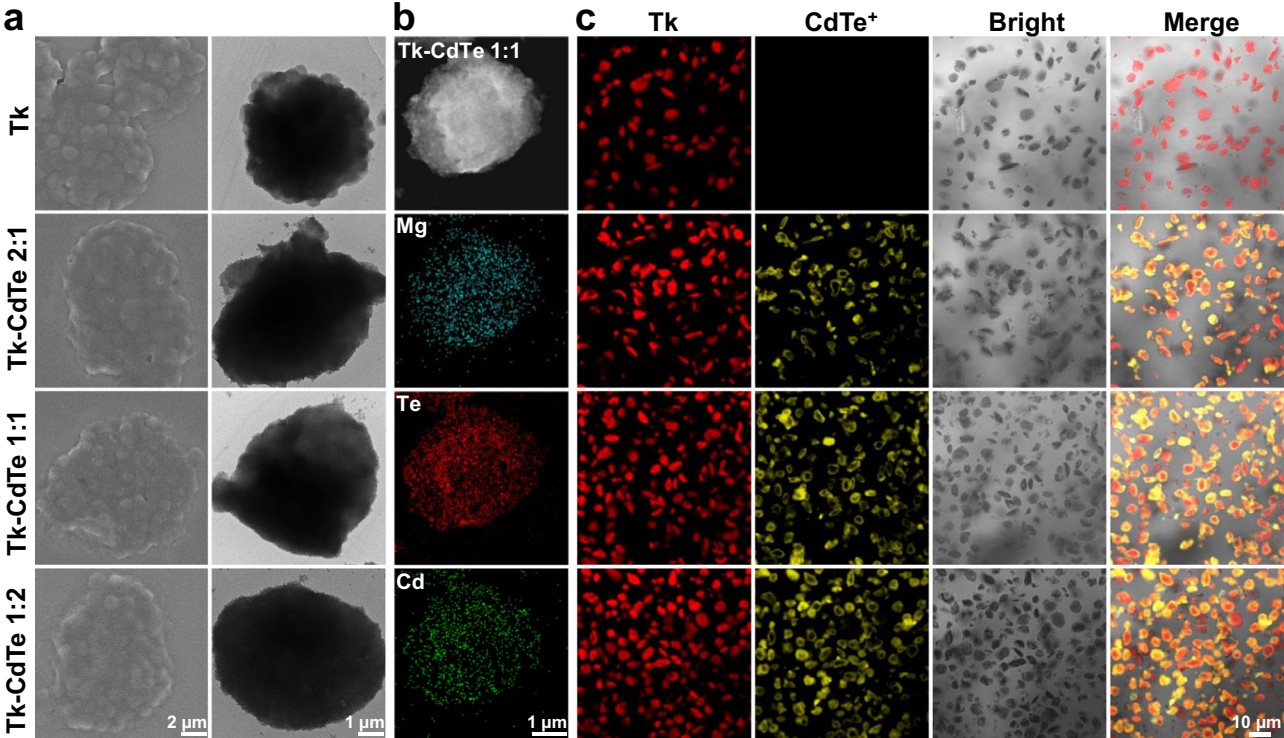

**Fig. 2 | Characterizations of thylakoid–CdTe energy modules. a** SEM (1st column) and TEM (2nd column) images of Tk, Tk–CdTe 2:1, Tk–CdTe 1:1 and Tk–CdTe 1:2. **b** EDS mapping images of the Tk–CdTe 1:1, in which blue, red and green signals indicate the Mg element from Tk, Te element and Cd element from CdTe, respectively. **c** CLSM images of TK (1st row), Tk–CdTe 2:1 (2nd row), Tk–CdTe 1:1 (3rd row) and Tk–CdTe 1:2 (4th row). Tk and CdTe are indicated by red and yellow fluorescence signals, respectively.

the luciferin/luciferase assay was employed to detect ATP production[2]. Amazingly, the combination of thylakoid with CdTe QDs resulted in remarkable enhancement in the regeneration activity for all the cofactors after illumination (Fig. 3). Note that there is no significant photothermal effect on cofactors regeneration under the light intensity of $0.1 W/cm^2$ (Supplementary Fig. 12).

NADPH is a cofactor that originally thylakoid can regenerate. As shown in Fig. 3a, although thylakoid and CdTe QDs could considerably regenerate NADPH under light illumination ($0.1 W/cm^2$) owing to their photocatalytic activity, the sample of Tk–CdTe 1:1 demonstrated an NADPH regeneration yield of $292.38 \pm 9.25 \mu M$ with $60 \mu g/mL$ chlorophyll (Chl) equivalent (note that Chl concentration was used to normalize the amounts of added thylakoid), 4.1- and 3.9-fold higher than those of pristine thylakoid and CdTe QDs, respectively. The NADPH regeneration yields over Tk and Tk–CdTe are positively correlated with the concentration of Chl equivalent in the range of 5 to $120 \mu g/mL$, while the yields have no sharp growth as the concentration of Chl equivalent reaches $60 \mu g/mL$. This slow growth is most likely ascribed to the limited $NADP^+$ concentration and illumination time[2]. Moreover, increasing the loading amount of CdTe QDs on thylakoid can further improve the NADPH regeneration efficiency (e.g., Tk–CdTe 2:1 *vs.* Tk–CdTe 1:1). In particular, Tk–CdTe 1:2 as energy module shows no enhancement for NADPH regeneration in contrast to Tk–CdTe 1:1, which should be caused by the limited amount of electron transfer medium in thylakoid or the hindered light absorption by over decoration of CdTe QDs[12]. Differently, NADH is a cofactor that pristine thylakoids can hardly regenerate. As displayed in Fig. 3b, only the yield of $18.05 \pm 1.64 \mu M$ was observed for the NADH regeneration by thylakoid with $120 \mu g/mL$ Chl equivalent. In comparison, CdTe QDs exhibited a higher NADH regeneration yield ($59.31 \pm 2.09 \mu M$), whereas the generated species are not bioactive which will be discussed in the following section. After thylakoid was coupled with CdTe QDs, the

regeneration of bioactive NADH can be substantially promoted. Of note, as the content of CdTe QDs increased, the performance of Tk–CdTe in NADH regeneration exhibited a volcano profile ranging from $4.68 \pm 0.98$ to $182.22 \pm 3.08 \mu M$ (Fig. 3b). The effects of Chl concentration and CdTe QDs loading amount on NADH regeneration yields are similar to that for NADPH regeneration. Furthermore, the presence of PDADMAC has no significant contribution to cofactors regeneration (Supplementary Fig. 13).

The light-driven regeneration of ATP confronts another different situation. The photophosphorylation in photosynthesis by thylakoid enables ATP regeneration, while CdTe QDs cannot offer the capability of regenerating ATP under light illumination ($0.1 W/cm^2$). Nevertheless, the combination of thylakoid with CdTe QDs significantly promoted the ATP regeneration as shown in Fig. 3c, also exhibiting a volcano trend with CdTe loading amount. Specifically, Tk–CdTe 1:1 gives the highest ATP production yield of approximately $410 \mu M$ with 10 min irradiation, which is 1.6 times higher than that by pristine thylakoid. Furthermore, the ATP regeneration yields are also positively correlated with the concentration of thylakoid (i.e., Chl equivalent) in the range of 5 to $120 \mu g/mL$, with the ATP yields ranging from $85.48 \pm 6.69$ to $412.89 \pm 13.11 \mu M$. Notably, when increasing the loading amount of CdTe QDs, the performance of Tk-CdTe in both NADH and ATP regeneration exhibited a volcano profile. This feature is most likely due to the limited amount of electron transfer medium in thylakoid or the hindered light absorption by over-decoration of CdTe QDs. Collectively, the results above demonstrated that the Tk–CdTe energy modules, which can efficiently harvest photon energy to generate photoexcited charges, dramatically enhanced the regeneration capability for NADPH, NADH and ATP cofactors as compared with pristine thylakoid and CdTe QDs under light illumination. According to the observed trends of activity variation, Tk–CdTe 1:1 has been identified as the most potent couple. It is worth pointing out that the yields

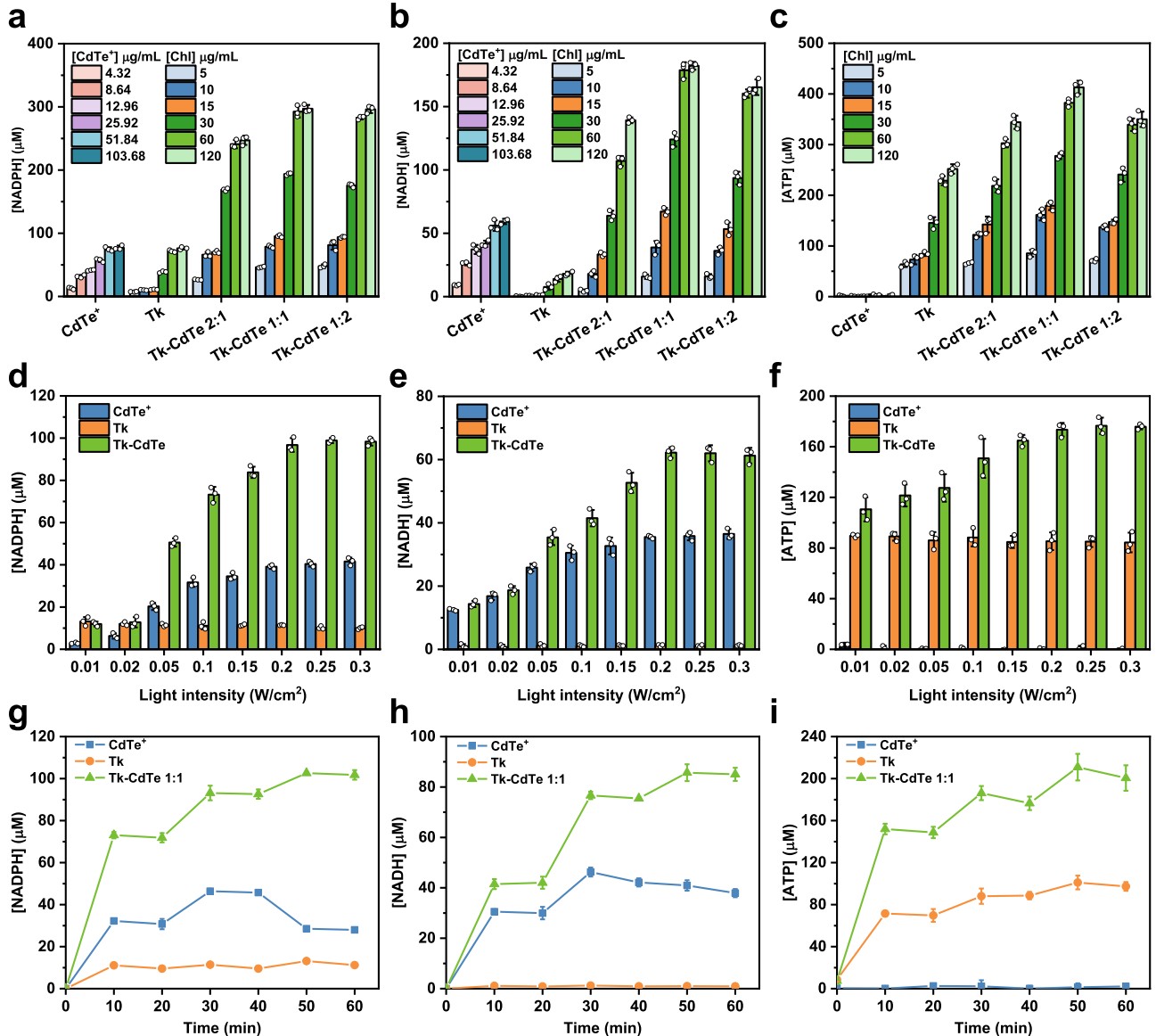

**Fig. 3 | Light-driven regeneration of NADPH, NADH and ATP cofactors by thylakoid−CdTe energy modules. a** NADPH, **b** NADH and **c** ATP regeneration by CdTe[+], Tk or Tk−CdTe with different concentrations. Note that Chl concentration was used to normalize the amounts of added thylakoid. The amounts of added CdTe[+] were calculated according to loading efficacy. The light intensity was 0.1 W/cm². Data points are reported as mean ± standard deviation derived from 3 independent experiments (n = 3). **d** NADPH, **e** NADH and **f** ATP regeneration under different light intensities by CdTe[+], Tk or Tk−CdTe 1:1 with 10 μg/mL Chl equivalent.

The amount of added CdTe[+] was 8.64 μg/mL. Data points are reported as mean ± standard deviation derived from 3 independent experiments. (n = 3). **g** NADPH, **h** NADH and **i** ATP production by CdTe[+], Tk or Tk−CdTe 1:1 with 10 μg/mL Chl equivalent through light−dark cycles. The amount of added CdTe[+] was 8.64 μg/mL. Data points are reported as mean ± standard deviation derived from 3 independent experiments (n = 3). The light intensity was 0.1 W/cm². The following experiments were carried out with the light intensity of 0.1 W/cm² unless specified otherwise. Source data are provided as a Source Data file.

of NADPH, NADH and ATP in light-driven cofactor regeneration can be tuned by adjusting chlorophyll (Chl) doses (Fig. 3a−c). An increase even to 10-fold in the yield of cofactor regeneration can be achieved by increasing the Chl concentration. Moreover, although the reduction of NAD[+] to NADH probably occurs at the same ferredoxin-NADP[+] reductase (FNR) for NADPH regeneration in Tk (Supplementary Fig. 14), Tk commonly displays significantly higher reaction activity for NADPH regeneration attributed to the high specificity of FNR toward NADP[+][20,21]. The control experiments eliminated the influence of possible impurities (e.g., plasma membrane, broken liposome and protein/proteoliposome) on NADH regeneration (Supplementary Fig. 15).

In the regeneration of multiple cofactors, light illumination is a highly important energy input. Our parallel experiments showed that, except ATP, there were negligible yields of NADPH and NADH

regeneration in the absence of light illumination (Supplementary Fig. 16). Previous reports indicated that the regeneration yields of cofactors in photoenzymatic processes were limited by light saturation[7,12]. Consistently, the cofactor regeneration capability of thylakoid in our work is not significantly influenced by the increasing light intensity (Fig. 3d−f). Intriguingly, we found that the regeneration yields of cofactors by Tk−CdTe were promoted by increasing light intensity, although pristine thylakoid exhibited a similar tendency of light-dependent cofactors regeneration to the literature[12]. As shown in Fig. 3d, the enhancement of NADPH regeneration was thoroughly quantified as a function of the light intensity. The production yields of NADPH were elevated from 11.95 ± 1.12 to 98.34 ± 1.85 μM by increasing light intensity at 0.01−0.3 W/cm². Similarly, the NADH or ATP regeneration was also enhanced with increasing light intensity (Fig. 3e, f).

The regeneration yields of NADPH and NADH by Tk−CdTe show a nearly monotonic enhancement with increasing light intensity at 0.01–0.2 W/cm², suggesting that charge carriers generation should be the rate-limiting step[22]. However, when the light intensity is increased to 0.2 W/cm², there is no obvious enhancement in cofactor regeneration, which is most likely due to the self-protection mechanism (i.e., cyclic electron flow) in the photosynthetic system of thylakoid[23]. Notably, as the light intensity varied, the sample of CdTe QDs showed a consistent tendency with Tk−CdTe despite its inferior performance (Fig. 3d, e). Impressively, the production yields of cofactors were found higher than other artificial systems[24,25]. These results demonstrate that the coupling of thylakoid and CdTe QDs endowed the hybrid energy modules with enhanced cofactor regeneration performance.

To better demonstrate the light-driven cofactors regeneration process, we examined the illumination response of Tk−CdTe with several light–dark cycles. Figure 3g–i shows that the cofactors can be regenerated and accumulated during the light illumination stage. In the dark stage, the regenerated cofactors were maintained. Remarkably, the regeneration of cofactors by Tk−CdTe was operable and stable over multiple light–dark cycles. In sharp contrast, in the case of CdTe QDs, the concentrations of cofactors decreased as the light–dark cycles proceeded up to 30 min, likely due to the aggregation of CdTe QDs[26] (Supplementary Figs. 17 and 18). Meanwhile, the morphology and dispersibility of thylakoid and Tk-CdTe have no obvious change after illumination (Supplementary Figs. 17 and 18). In literature, to enhance the photoactivity of thylakoid, external ferredoxin (Fdx) and ferredoxin-NADP⁺ reductase (FNR) were usually added as supplementary to increase the rate of electron transfer[2,12,27]. Amazingly, our Tk−CdTe, without the need for any external supplements, demonstrated comparative capability to the enhanced performance of thylakoid by adding external Fdx and FNR[2,12]. Given the high cost of external protein separation, commercial enzymatic transformations were not economically feasible[28]. As such, our Tk−CdTe energy modules with potential cost saving will be beneficial to the economic viability and feasibility of a practical application.

## Light-driven CO$_2$ conversion

The efficient regeneration of multiple cofactors offers the feasibility of working together with CO$_2$ convertases for light-driven CO$_2$ conversion. Prior to light-driven CO$_2$ conversion measurements, we first examined the cofactor regeneration mechanism. Our fluorescence spectroscopy characterization revealed that the photoemission of CdTe QDs was significantly quenched by interfacing with thylakoid (Fig. 4a, b and Supplementary Fig. 19), suggesting that the photoexcited electrons in CdTe were transferred to thylakoid suppressing radiative charge recombination. This finding was also observed in transient fluorescence spectroscopy by analyzing lifetime values[29] (Fig. 4c, d, Supplementary Fig. 20, Supplementary Table 3). The positively charged CdTe⁺ QDs exhibit a comparable average lifetime (27.95 ns) in contrast to pristine CdTe QDs (24.35 ns). After combination with thylakoid, the lifetime of CdTe QDs decreases notably to 7.60 ns. Correspondingly, after combination with CdTe QDs, the lifetime of thylakoid (i.e., Chl) slightly increases from 0.76 to 1.08 ns (Fig. 4d, Supplementary Table 4)[30,31]. These results confirm that the photoexcited electrons in CdTe transfer to thylakoid. To further confirm this electron transfer behavior, we investigated the system of Tk−CdTe using 2,6-dichlorophenolindophenol (DCPIP) as a probe. The DCPIP dye is an artificial electron acceptor that can capture the electron transfer from photosystem II (PSII) to photosystem I (PSI) in photosynthesis (Supplementary Fig. 21)[32]. As shown in Fig. 4e, under light illumination, the absorbance of DCPIP gradually decreased by prolonging illumination time in the case of pristine thylakoid, implying that the photoexcited electrons from thylakoid enable DCPIP reduction. For the Tk−CdTe 1:1, the reduction of DCPIP was obviously accelerated in comparison to thylakoid. As CdTe QDs alone could not reduce DCPIP, the accelerated reduction of DCPIP should be attributed to the supply of photoexcited electrons from CdTe QDs to thylakoid.

Considering that photosystem II (PSII) in thylakoid plays a key role in transferring protons and electrons, accompanied by cofactors generation in natural photosynthesis[33], we then seek to unveil whether PSII underlies the accelerated electrons transfer in our Tk−CdTe module. To this end, PSII was isolated from thylakoid[34] and combined with CdTe QDs (Supplementary Fig. 22, Supplementary Table 5), denoted as PSII-CdTe, which showed superior electron transfer to Tk-CdTe. As shown in Fig. 4f, after combination with PSII, the lifetime of CdTe⁺ QDs shows a much faster decay (1.61 ns, Supplementary Table 6) compared to that in Tk-CdTe. Meanwhile, after combination with CdTe QDs, the lifetime of PSII increases from 0.86 to 1.48 ns (Fig. 4g and Supplementary Table 7). These results confirm the electron transfer from the excited CdTe QDs to PSII. Furthermore, the efficiency of DCPIP reduction by PSII-CdTe was superior to that by Tk-CdTe (Fig. 4h), which is most likely ascribed to the direct contact of CdTe QDs with PSII. Therefore, the effective adsorption sites for CdTe⁺ on thylakoid membrane are probably the negatively charged carboxyl in photosynthetic protein (Supplementary Fig. 8 and 22), and the interface between CdTe QDs and PSII plays a key role in electron transport, which increases the supply of photogenerated electrons to boost the cofactors regeneration by Tk. As the conduction band (CB) of PSII (light-harvesting P680) is −0.58 V (vs. NHE)[35,36] and the CB of CdTe QDs was measured to be −1.5 V (vs. NHE)[37–39], the photoexcited electrons in CdTe QDs can be favorably injected into the CB of PSII through an interfacial charge-transfer process, subsequently transferred to PSI for cofactors regeneration via electron transport chain, and ultimately consumed by CO$_2$ reductase (Fig. 4i and Supplementary Fig. 23). Meanwhile, the photogenerated holes in CdTe QDs will be consumed by oxidation of sodium L-ascorbate in the reaction buffer, while the photogenerated holes by P680 are likely finally consumed for oxidization of water in PSII. The results collected above suggest that PSII could play an essential role in mediating electron transfer between CdTe QDs and thylakoid.

To further explore whether our thylakoid-based biotic−abiotic energy module that promotes cofactors regeneration is available to other quantum dots, the well-studied colloidal semiconductor quantum dots, such as CdS[40] and MoS$_2$ QDs[41], were applied to construct the hybrid energy modules for cofactors regeneration (Supplementary Figs. 24 and 25, Supplementary Tables 8 and 9). Both CdS and MoS$_2$ QDs exhibit good dispersibility, and the integrity of thylakoid structure can be well maintained after incorporation of CdS or MoS$_2$ QDs. Moreover, CdS and MoS$_2$ QDs also exhibit fluorescence quenching effects after combination with thylakoid. However, not only CdS and MoS$_2$ QDs but also their hybrids with thylakoid have a remarkably lower capacity for cofactors regeneration in contrast to CdTe QDs and its hybrid (Supplementary Fig. 26). Such an obvious distinction for cofactors regeneration among the different QDs suggest the unique superiorities of CdTe QDs for constructing hybrid energy module. In principle, compared with CdTe (Supplementary Fig. 27), CdS and MoS$_2$ QDs own wide band gap and thus their light absorption is mainly in ultraviolet region[14,40,42] (Supplementary Figs. 24 and 25), and as such, less electrons can be excited from the CdS and MoS$_2$ QDs under the simulated solar light.

Given the enhanced cofactor regeneration through electron transfer, a critical issue for powering CO$_2$ convertases is the bioactivity of regenerated cofactors. Previous reports indicated that non-enzymatic regeneration of cofactors may suffer from irreversible formation of inactive dimer or isomer[43–46]. For this reason, we evaluated whether the cofactors regenerated by Tk−CdTe are naturally bioactive using ¹H nuclear magnetic resonance (NMR) spectroscopy. The ¹H NMR spectra for bioactive NADP⁺/NAD⁺ and NADPH/NADH should present characteristic signals at 9.28/9.32 and 6.95/6.94 ppm,

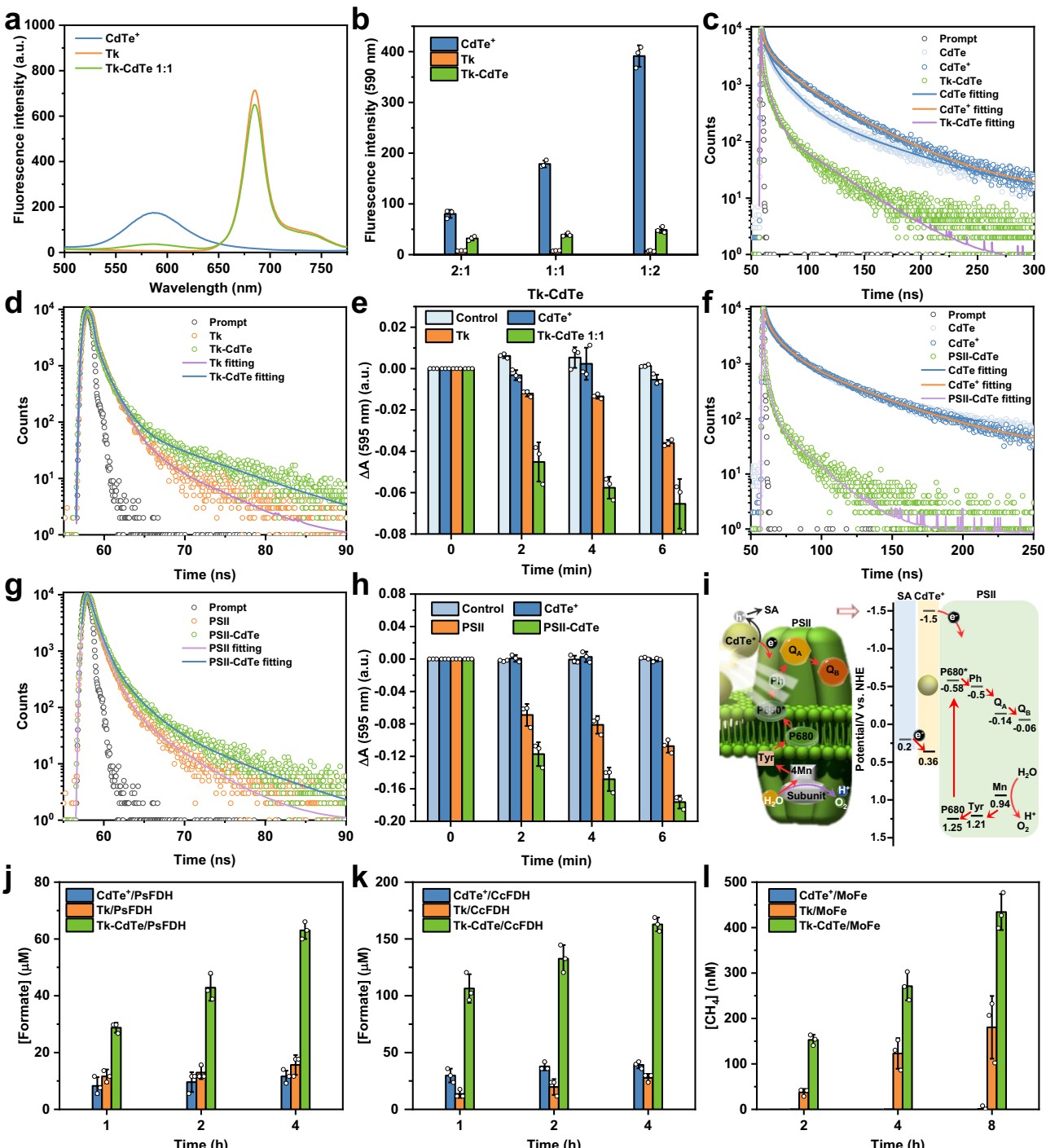

respectively[28,47]. Under light illumination, no clear characteristic signal at 6.95 was observed for the cofactors regenerated by pristine thylakoid or CdTe QDs (Supplementary Fig. 28). In sharp contrast, this characteristic signal of NADPH at 6.95 was unambiguously resolved for Tk−CdTe 1:1. Although CdTe QDs could present a certain signal for the regeneration of NADH in spectrophotometry measurement (Fig. 3b), the generated species are not bioactive, totally different from the case of Tk−CdTe 1:1. This finding is indicated by the result that the NADH regenerated by CdTe QDs presented quite low signals in ¹H NMR (Supplementary Fig. 29). The signal of CdTe QDs in spectrophotometry measurement can be ascribed to the increase of absorption intensity, caused by the formation of non-bioactive isomer (1,6-NAD(P)H) through bare CdTe QDs treatment[48,49]. Note that although thylakoid can produce bioactive NADPH, it requires a much higher

amount of Tk (60 μg/mL Chl equivalent) to produce detectable NADPH (Supplementary Fig. 30), which is remarkably inferior to Tk−CdTe.

To suppress undesirable side reactions in non-enzymatic cofactors regeneration, the electron mediator of pentamethylcyclopentadienyl rhodium bipyridine ([Cp*Rh(bpy)H₂O]²⁺, denoted as [M]) has been frequently used as hydride transfer species to exclusively form bioactive cofactors[50]. Although CdTe QDs alone could produce bioactive cofactor with [M] as a supplementary, it requires a high amount of CdTe QDs (51.84 μg/mL) and [M] (0.4 mM) to produce a similar amount of NADH compared with Tk−CdTe (Supplementary Fig. 31). Furthermore, we found that there was a negligible promotion for NADPH/NADH regeneration when the external [Cp*Rh(bpy)H₂O]²⁺ was added to the system of Tk−CdTe 1:1 (Supplementary Fig. 32). This suggests that our platform for multiple cofactors regeneration is

**Fig. 4 | Light-driven CO$_2$ conversion by integrating reductases with cofactors regeneration using Tk–CdTe. a** Fluorescence spectra of CdTe$^+$, Tk and Tk–CdTe 1:1 with an excitation wavelength of 400 nm. **b** Fluorescence intensity (590 nm) of CdTe$^+$, Tk and Tk–CdTe with an excitation wavelength of 400 nm. Data points are reported as mean ± standard deviation derived from 3 independent experiments ($n$ = 3). **c, d** Transient fluorescence spectra of **c** CdTe QDs, CdTe$^+$ and Tk–CdTe 1:1 with an emission wavelength of 590 nm and **d** Tk and Tk–CdTe 1:1 with an emission wavelength of 685 nm. "Prompt" represents the instrument response function. **e** The absorption changes of DCPIP at 595 nm in the presence of CdTe$^+$, Tk or Tk–CdTe 1:1. Control denotes that DCPIP was treated similarly but with phosphate-buffered saline (PBS). Data points are reported as mean ± standard deviation derived from 3 independent experiments ($n$ = 3). **f, g** Transient fluorescence spectra of **f** CdTe QDs, CdTe$^+$ and PSII-CdTe with an emission wavelength of 590 nm and **g** PSII and PSII-CdTe with an emission wavelength of 685 nm. **h** The absorption changes of DCPIP at 595 nm in the presence of CdTe$^+$, PSII or PSII–CdTe. Control denotes that DCPIP was treated similarly but with buffer E1. Data points are

reported as mean ± standard deviation derived from 3 independent experiments ($n$ = 3). **i** Schematic illustration of the electron transfer from CdTe$^+$ to PSII with their corresponding redox potentials. Mn, Tyr, P680, Ph, Q$_A$ and Q$_B$ are the abbreviations of the electron transfer factors in PSII. SA is the abbreviation of sodium L-ascorbate[57]. Note that the band gap and positions of CdTe$^+$ were calculated with UV–Vis diffuse reflectance spectra and linear sweep voltammetry (LSV). PSII is depicted based on previous reports[35,36,58,59]. **j** Formate production from CO$_2$ by NADPH regeneration coupled with NADPH-dependent PsFDH. Data points are reported as mean ± standard deviation derived from 3 independent experiments ($n$ = 3). **k** Formate production from CO$_2$ by NADH regeneration coupled with NADH-dependent CcFDH. Data points are reported as mean ± standard deviation derived from 3 independent experiments ($n$ = 3). **l** Methane production by ATP regeneration coupled with ATP-dependent MoFe. Data points are reported as mean ± standard deviation derived from 3 independent experiments ($n$ = 3). Source data are provided as a Source Data file.

---

independent of electron mediator, which avoids potential biotoxicity of metals derived from electron mediator. To further test the bioactivity toward regenerated NADPH, we used glyoxylate/hydroxypyruvate reductase (GhrA) from *Escherichia coli*, which can catalyze the NADPH-dependent reduction of glyoxylate to glycolate, to examine the system of Tk–CdTe 1:1 (Supplementary Figs. 33–35)[2]. The $^1$H NMR spectra for glyoxylate and glycolate presented characteristic signals at 5.05 and 3.93 ppm, respectively. After 30 min illumination, a substantially higher characteristic signal at 3.93 emerged with Tk–CdTe 1:1, indicating the superior performance for producing glycolate as compared with pristine thylakoid or CdTe QDs. Taken together, the results above reveal that Tk–CdTe offers the exclusive regeneration of bioactive cofactors with desirable quality, which is attributed to the combination of thylakoid with CdTe.

Based on the regeneration of multiple bioactive cofactors, Tk–CdTe can be coupled with various CO$_2$ reductases for photoenzymatic conversion. The NADPH-dependent formate dehydrogenase (PsFDH)[51] and NADH-dependent formate dehydrogenase (CcFDH)[52] were implemented in the conversion of CO$_2$ to formate (Supplementary Fig. 36), while the ATP-required nitrogenase (MoFe) remodeled with two amino acid substitutions was employed for the reduction of CO$_2$ to methane (CH$_4$)[53]. To identify the characters of these CO$_2$ reductases, enzymatic activity assays were first conducted under different conditions (Supplementary Figs. 37–39). As expected, PsFDH and CcFDH demonstrated NADPH- and NADH-dependent enzymatic reactions, respectively, which was confirmed by the optical intensity decrease after co-incubation. In the presence of sufficient cofactors, PsFDH, CcFDH and MoFe all exhibit concentration-dependent CO$_2$ reduction behavior, showing over 200% increase in product yield with protein concentration promotion.

Furthermore, photoenzymatic conversion of CO$_2$ was conducted based on our multiple cofactors regeneration platform. As shown in Fig. 4j–l, the system of coupling Tk-CdTe 1:1 with enzyme (namely, Tk–CdTe/enzyme) produced a dramatically higher concentration of desired products than the coupling system with pristine thylakoid or CdTe QDs when sodium L-ascorbate was used as the electron donor. In detail, after 4 h light illumination, Tk–CdTe/PsFDH system produced 63.02 ± 3.02 μM formate, exceeding the production by Tk/PsFDH and CdTe$^+$/PsFDH by 420 ± 90% and 540 ± 26%, respectively (Fig. 4j). Under light illumination, the formate production in Tk–CdTe/PsFDH system is time-dependent, consistently yielding higher amounts than its counterparts. Similar results were achieved in the Tk–CdTe/CcFDH system, yielding 162.82 ± 6.05 μM formate within 4 h (Fig. 4k). As MoFe reductase was coupled with Tk–CdTe, the photoenzymatic CO$_2$ conversion system can be turned into CH$_4$ production. Similarly to the cases of formate production, the photoenzymatic CH$_4$ production in the Tk–CdTe/MoFe system well exceeded those using Tk/MoFe and CdTe$^+$/MoFe (Fig. 4l). The products of formate and methane both

require a proton-coupled electron transfer process for their formation so that the enhanced electron supply by Tk–CdTe can significantly facilitate the process. The information gleaned above proves that the multiple cofactors, regenerated by our designed energy modules, are enzymatically active for coupling with various CO$_2$ convertases for customizable products.

## Programmable artificial photosynthetic cells

The results above demonstrated that the native energy machinery of photosynthesis can be engineered by a synthetic component, providing a platform that wondrously enhances the performance of components. The platform is functionally coupled with enzymes in an integrated fashion, using light energy to produce value-added compounds from CO$_2$. To overcome the limitations of individual bulk experiments (e.g., inefficient cofactor shuttling, severe self-shading effects, and the limited number of parallel assays)[2], a miniaturized and confined environment was further explored for our platform[54]. We hence implemented the photoenzymatic conversion of CO$_2$, essentially based on the multiple cofactor regeneration platform, inside artificial photosynthetic cells. Based on microfluidics (Supplementary Fig. 40), we encapsulated the representative Tk–CdTe 1:1 into water-in-oil microdroplets stabilized by a block-copolymer surfactant, forming artificial photosynthetic cells[2]. The typical workflow is displayed in Fig. 5a, in which a microfluidic chip was fixed on the objective table for analysis in real time. Microdroplets with -100 μm in diameter and distinctive rough edge were generated by a flow-focusing junction fabricated in a polydimethylsiloxane (PDMS) chip. The reasonable composition was visualized by confocal microscopy, indicating that the desired artificial photosynthetic cells were successfully constructed (Fig. 5b, Supplementary Fig. 41 and Movie 1).

Upon the fabrication of artificial cells, we studied cofactor regeneration activity by assessing NADH fluorescence in the cells (Supplementary Figs. 41–43). As anticipated, the artificial cells with Tk–CdTe can efficiently regenerate NADH judging from the fluorescence (Fig. 5c). Although encapsulating bare CdTe QDs or thylakoid could also make up the cells, they exhibited feeble or negligible fluorescence after 10 min illumination in sharp contrast to the cells with Tk–CdTe (Fig. 5c and Supplementary Fig. 44). Furthermore, we observed that the NADH regeneration rate by Tk–CdTe in the cells was 2.08 μM per μg/mL of Chl equivalent after 10 min illumination, significantly higher than other counterparts and comparable to the rates obtained earlier in bulk experiments (Figs. 3b and 5d). This outcome is indicative of the regeneration activity by Tk–CdTe energy modules.

After creating the artificial photosynthetic cells for cofactors regeneration, we ultimately aimed at cofactors-dependent photoenzymatic reduction of CO$_2$. Thus, we next tested the capability of the artificial photosynthetic cells containing Tk–CdTe to power individual enzyme reactions through co-encapsulation of enzymes and

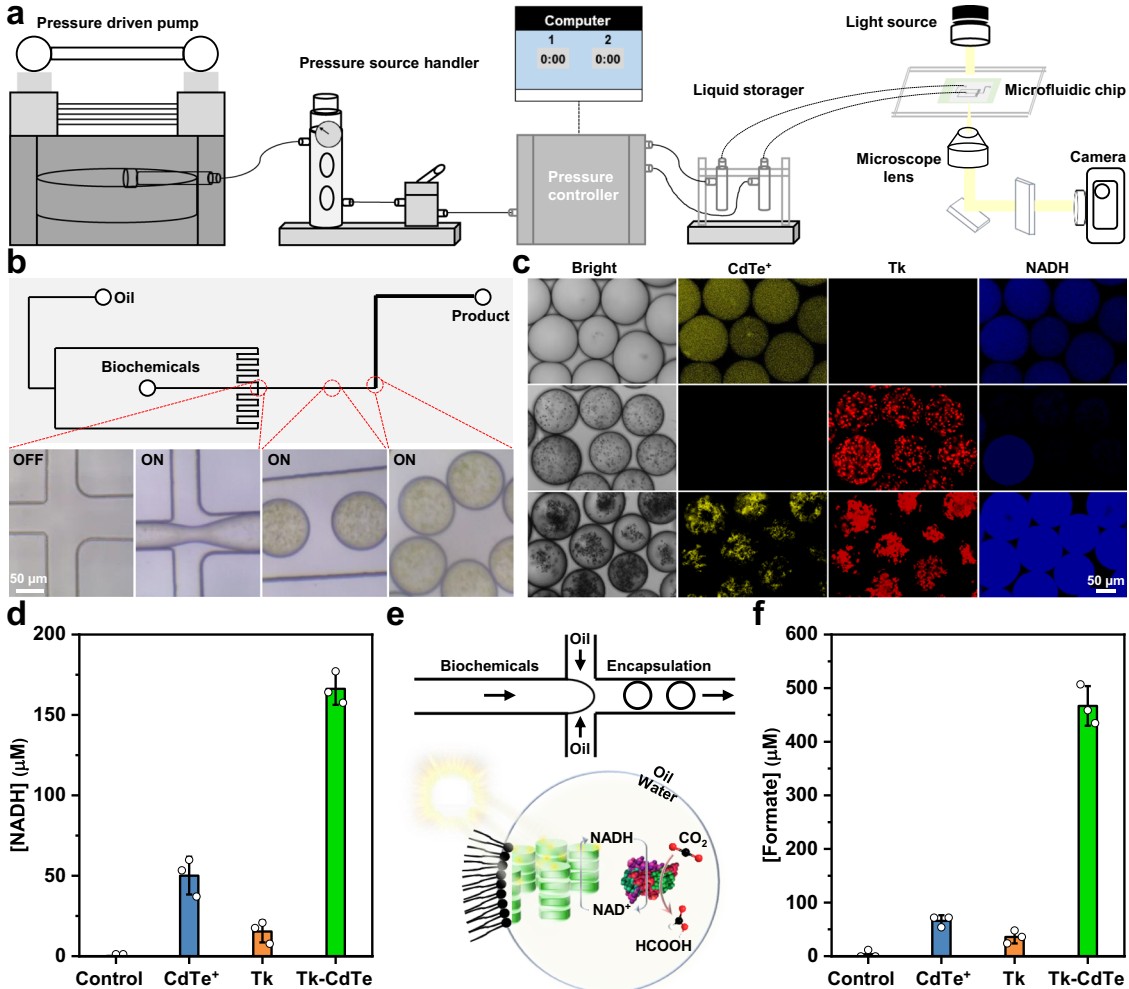

**Fig. 5 | Light-driven NADH regeneration and CO₂ conversion in artificial photosynthetic cells. a** Schematic illustration of a microfluidic device for artificial cell fabrication and real-time observation. **b** Fabrication process of Tk–CdTe-based artificial cells for NADH regeneration in a microfluidic chip. **c** CLSM images of NADH regeneration in artificial photosynthetic cells (top row for CdTe⁺, middle row for Tk, and bottom row for Tk–CdTe) after 10 min light illumination. Regenerated NADH was detected with an excitation wavelength of 405 nm. **d** NADH regeneration by CdTe⁺, Tk or Tk–CdTe with 80 μg/mL Chl equivalent in artificial photosynthetic cells after 10 min light illumination. Control denotes the counterpart only containing buffer E. The amount of added CdTe⁺ was 69.12 μg/mL. Data points are reported as mean ± standard deviation derived from 3 independent experiments (n = 3). **e** Schematic illustration of CO₂ conversion into formate by NADH-dependent CcFDH in artificial photosynthetic cells. **f** Formate production from CO₂ by CdTe⁺, Tk or Tk–CdTe in artificial photosynthetic cells after 1 h light illumination. Control denotes the counterpart only containing buffer G and CcFDH. Data points are reported as mean ± standard deviation derived from 3 independent experiments (n = 3). Source data are provided as a Source Data file.

substrates (Fig. 5e, Supplementary Figs. 45 and 46 and Movie 2). Indeed, as shown in Fig. 5f, production of ~460 μM formate with CcFDH was observed after 1 h illumination whereas other counterparts exhibited substantially weaker activity. This artificial photosynthetic cell shows a superior performance (reaction rate: 201.84 ± 23.89 μM mg⁻¹ h⁻¹ and internal quantum efficiency[55] (IQE): 2.46 ± 0.19%) compared to the state-of-the-art studies on the cofactor-dependent photochemical CO₂ conversion (Supplementary Table 10). Meanwhile, in the parallel experiments, NADH was regenerated at distinct amounts which can be correlated to different counterparts (Supplementary Fig. 47), demonstrating the excellent performance of Tk–CdTe that distinctly exceeds its components. Apparently, our artificial photosynthetic cells can also be implemented in NADPH-dependent PsFDH and ATP-required MoFe reductases for conversion of CO₂ to formate and methane, respectively (Supplementary Figs. 48–56 and Movie 3). Overall, these results demonstrate that our designed Tk–CdTe energy modules can work together with CO₂ convertases and multiple cofactors in a cell-like environment, converting

CO₂ into selective products in a programmable manner. Such a successful connection of natural modules with synthetic components in cell-sized compartments creates photosynthetic entities for customizable CO₂ conversion.

## Discussion

We have prepared the thylakoid–quantum dot hybrid system as a highly efficient energy module for multiple cofactors regeneration, realizing multifarious photoenzymatic CO₂ conversion. Without external supplements, the regenerative NADPH, NADH and ATP are all bioactive cofactors. In particular, the regeneration of these cofactors is substantially promoted by the enhanced electron transfer in Tk–CdTe. As such, this biotic–abiotic hybrid platform can be further coupled with NADPH-dependent PsFDH, NADH-dependent CcFDH and ATP-dependent MoFe, converting CO₂ into HCOOH and CH₄, respectively. By encapsulating and operating the platform in cell-sized droplets, we have developed artificial photosynthesis cells to achieve controllable CO₂ conversion by integration with various reductases. This work

provides a concept for rationally designing artificial cells de novo, which essentially are programmable through enzyme selection for customizable photosynthesis.

## Methods

### Preparation of Tk−CdTe

CdTe QDs were synthesized and subsequently functionalized with PDADMAC to obtain the positively charged CdTe+ QDs, and thylakoids were prepared by isolating them from young spinach. The modification of thylakoid with CdTe+ was accomplished by mixing the as-prepared thylakoid with CdTe+ (mass ratio of thylakoid (Chl) to CdTe+ was 2:1, 1:1 and 1:2, respectively) in a vial with gently stirring for 10 min, followed by centrifugation at 3000 × g for 10 min at 4 °C to remove unbounded CdTe+, which yielded the CdTe+-modified thylakoid (namely, Tk−CdTe 2:1, Tk−CdTe 1:1 and Tk−CdTe 1:2, respectively). The same method was used to obtain Tk-CdS and Tk-MoS$_2$ (Supplementary Tables 8 and 9).

### NADPH, NADH and ATP production assays

NADPH or NADH regeneration activity was assayed by monitoring the absorption at 340 nm in a 1 mL reaction volume. The reaction buffer contained 50 mM 4-(2-hydroxyethyl)piperazine-1-ethanesulfonic acid (HEPES)-KOH pH 7.8, 5 mM K$_2$HPO$_4$, 3 mM NADP+, 10 mM sodium L-ascorbate, 10 mM KCl, 5 mM MgCl$_2$ and samples of various concentrations, which was illuminated with white light (AM 1.5 filter, sunlight simulation) at 0.1 W/cm$^2$ for 10 min. The resulting reaction buffer was centrifuged at 3000×g for 10 min at 4 °C. The resultant supernatant was then subjected to a spectrophotometer (Metash, V-5000). ATP regeneration activity was measured by using the ATP assay kit. The reaction buffer contained 50 mM HEPES-KOH pH 7.8, 3 mM ADP, 5 mM K$_2$HPO$_4$, 10 mM sodium L-ascorbate, 10 mM KCl, 5 mM MgCl$_2$ and samples of 10 μg/mL Chl equivalent (or equal amount of CdTe+, i.e., Chl equivalent × loading efficacy), which was illuminated with white light (AM 1.5 filter, sunlight simulation) at 0.1 W/cm$^2$ for 10 min. The resulting reaction buffer was centrifuged at 3000×g for 10 min at 4 °C. The resultant supernatant was then subjected to an ATP assay kit. The regeneration of ATP was determined by luminescence recorded with a multifunctional microplate reader (Spectramax M3, Molecular Devices).

To determine how light intensity affects NADPH, NADH and ATP production, a similar regeneration assay was performed as described above but illuminated with white light at different intensities for 10 min. To determine how illumination time affects NADPH or NADH production, a similar regeneration assay was performed as described above but illuminated with white light at 0.1 W/cm$^2$ for different times. To determine how illumination affects NADPH, NADH and ATP production, a similar regeneration assay was performed as described above but without illumination. The reported values are averages of triplicate measurements. The error bars represent standard deviations.

### Assays for electron transfer from PSII to PSI

The dispersion of Tk−CdTe 1:1 in PBS was added into DCPIP solution, to achieve a final Chl equivalent of 20 μg/mL (17.28 μg/mL for CdTe+) and DCPIP concentration of 100 μM. The resulting mixture was subsequently illuminated with white light (AM 1.5 filter, sunlight simulation) at 0.1 W/cm$^2$ for a certain time (0, 2, 4 and 6 min, respectively). The resulting reaction buffer was then centrifuged at 3000×g for 10 min at 4 °C. The resultant supernatant was then subjected to measurement of absorption at 595 nm with a microplate reader (CMax Plus, Molecular Devices). Controls were assayed similarly but only with DCPIP (100 μM) in PBS. The reported values are averages of triplicate measurements. The error bars represent standard deviations.

### Enzymatic activity assays

The details for cloning, expression and purification of GhrA, PsFDH and CcFDH, as well as cloning, expression and crude extract of MoFe, are provided in Supplementary Methods. GhrA was first evaluated by

monitoring the decrease of absorption at 340 nm in a 0.6 mL reaction volume. The reaction involved 50 mM HEPES-KOH pH 7.8, 5 mM K$_2$HPO$_4$, 0.25 mM NADPH, 10 mM KCl, 5 mM MgCl$_2$ and 1.5 mM glyoxylate, and was initiated by the addition of GhrA (6 μg/mL, final concentration). After 10 min reaction, the absorbance at 340 nm was measured. Meanwhile, to explore the glycolate production during the process above, the resultant mixture (0.5 mL) was then subjected to $^1$H NMR spectrum recorded with Bruker AVANCE AV III 400. Controls were assayed similarly but only with glyoxylate or glycolate. PsFDH was evaluated by monitoring the decrease of absorption at 340 nm in a 1 mL reaction volume. The reaction involved PBS (pH 7.4), 0.2 mM NADPH and PsFDH (0.25 mg/mL), which was purged with N$_2$ for 10 min and initiated with the infusion of CO$_2$. After 30 min reaction, the absorbance at 340 nm was measured. CcFDH was evaluated with a similar assay but without N$_2$ and CO$_2$ purging. The reaction solution contained sodium phosphate buffer (PB, 100 mM, pH 7.0), 0.1 mM NADH, 1 mM NaHCO$_3$ and CcFDH (0.125 mg/mL).

The CO$_2$ reduction activity of PsFDH was measured in PBS (1 mL, pH 7.4) at 25 °C. The reaction mixtures (PBS, 4 mM NADPH and PsFDH) were purged with N$_2$ for 10 min, and the reaction was initiated by the infusion of CO$_2$, followed by incubation at 25 °C for 2 h. CO$_2$ reduction activity of CcFDH was measured in PB (0.5 mL, 100 mM, pH 7.0) at 37 °C. The reaction mixtures (PB, 4 mM NADH, 10 mM NaHCO$_3$ and CcFDH) were incubated at 37 °C for 1 h. The product was estimated instantaneously by following the Lang and Lang method[56]. Then, 100 μL of reaction mixtures were mixed with 200 μL of solution A, 10 μL of solution B, and 700 μL of solution C, followed by incubation at 50 °C for 1.5 h with vibrating once. The resultant was centrifuged (12,000×g, 10 min), and the obtained suspension was measured at 515 nm with a spectrophotometer (Metash, V-5000). Next, 0.5 g of citric acid and 10 g of acetamide were dissolved in isopropanol (100 mL) to prepare solution A; 30 g of sodium acetate was dissolved in water (100 mL) to prepare solution B; solution C was 100% acetic anhydride. Sodium formate was used for standard calibration. The reported values are averages of triplicate measurements. The error bars represent standard deviations.

The CO$_2$ reduction activity of MoFe was measured in photo-enzymatic reaction buffer (2 mL in 35-mL quartz tube) at 25 °C. The reaction involving 50 mM HEPES-KOH pH 7.8, 5 mM K$_2$HPO$_4$, 50 mM ATP, 10 mM sodium L-ascorbate, 10 mM KCl, 5 mM MgCl$_2$, 10 mM sodium dithionite and MoFe crude extracts was initiated by the infusion of CO$_2$, followed by incubation at 25 °C for 8 h. The product was quantified by gas chromatography (GC, 7890B, Ar carrier, Agilent) equipped with a flame ionization detector (FID) and thermal conductivity detector (TCD).

### Photoenzymatic reduction of CO$_2$ to formate and methane

Photoenzymatic reduction of CO$_2$ to formate was carried out under 25 °C in a quartz tube with white light (AM 1.5 filter, sunlight simulation, 0.1 W/cm$^2$) illumination. The reaction involving 50 mM HEPES-KOH pH 7.8, 5 mM K$_2$HPO$_4$, 10 mM sodium L-ascorbate, 10 mM KCl, 5 mM MgCl$_2$, 4 mM NADP+ or NAD+, 1 mg/mL PsFDH or 0.5 mg/mL CcFDH and samples of 15 μg/mL Chl equivalent (12.96 μg/mL for CdTe+) was initiated by the infusion of CO$_2$ for 10 min. After light illumination, the resulting reaction buffer was centrifuged at 3000×g for 10 min at 4 °C. The formate concentration in the resultant supernatant was estimated instantaneously by following the Lang and Lang method[56]. To accurately determine the concentration of CO$_2$ during the reaction process, NaHCO$_3$ (10 mM) was used as a substrate instead of CO$_2$ infusing. Photoenzymatic reduction of CO$_2$ into formate based on NaHCO$_3$ was conducted as described above.

Photoenzymatic reduction of CO$_2$ to methane was carried out under 25 °C in a quartz tube with white light (AM 1.5 filter, sunlight simulation, 0.1 W/cm$^2$) illumination. The reaction involving 50 mM HEPES-KOH pH 7.8, 5 mM K$_2$HPO$_4$, 100 mM ADP, 10 mM sodium L-

ascorbate, 10 mM KCl, 5 mM MgCl$_2$, 100 mM sodium dithionite, 8 mg/mL MoFe crude extracts and samples of 15 μg/mL Chl equivalent (12.96 μg/mL for CdTe$^+$) was initiated by the infusing of CO$_2$ for 10 min, followed by illumination. The concentration of methane was quantified by gas chromatography (GC, 7890B, Ar carrier, Agilent) equipped with a flame ionization detector (FID) and thermal conductivity detector (TCD).

## Fabrication of artificial photosynthetic cells

Artificial photosynthetic cells were fabricated by microfluidic devices. In typical experiments, the microfluidic channels contained one oil inlet and one aqueous inlet. For the oil phase, fluorinated oil containing 2% surfactant was used. For the aqueous phase, biochemicals in the buffer were pumped (see the components in Supplementary Table 11). Droplets were co-flown with the oil phase (~90 mbar) and aqueous phase (~60 mbar), and the resultant emulsions were then collected in a tube at the outlet, which yielded artificial photosynthetic cells. To obtain compact droplets in fluorinated oil, generated emulsions were stabilized for 5–10 min before being collected.

## CO$_2$ conversion in artificial photosynthetic cells

For CO$_2$ conversion with CcFDH, artificial photosynthetic cells were created from "Fabrication of artificial photosynthetic cells", and the collected artificial photosynthetic cells with a total volume of 400 μL were illuminated with white light (AM 1.5 filter, sunlight simulation) at 0.1 W/cm$^2$ for 1 h. Subsequently, 150 μL of fluorinated oil was added to 90 μL of pre-treated artificial photosynthetic cells. Then, 60 μL of 1H, 1H, 2H, 2H-perfluoro-1-octanol was then added, and the resultant mixture was vortexed and centrifuged. The aqueous phase was pipetted and quenched with 1% of HCl. The formate concentration in the resultant supernatant was estimated instantaneously by following the Lang and Lang method[56].

For CO$_2$ conversion with PsFDH and MoFe, artificial photosynthetic cells were created from "Fabrication of artificial photosynthetic cells" with slight modification. Briefly, the pressure-driven pump was replaced with a compressed-gas steel cylinder, and the microfluidic channels were gently infused with argon for 10 min before droplets were formed. The droplets were co-flowed with the oil phase and aqueous phase with CO$_2$ driving, and the resultant emulsions were then collected in a tube at the outlet, which yielded artificial photosynthetic cells. To obtain compact droplets in fluorinated oil, the generated emulsions were stabilized for 5–10 min before being collected. The collected artificial photosynthetic cells with a total volume of 1 mL were illuminated with white light (AM 1.5 filter, sunlight simulation) at 0.1 W/cm$^2$ for 4 h. The concentration of formate or methane was quantified as described above.

## Statistics and reproducibility

All the statistical analyses were performed using Origin 8.0, Excel 2019, and Nano Measurer software. Statistical comparisons were carried out by performing two-sided Student's $t$-test. For gel or micrographs, the reported results have been checked for consistency with 3 individual experiments.

## Reporting summary

Further information on research design is available in the Nature Portfolio Reporting Summary linked to this article.

## Data availability

The authors declare that all data supporting the findings of this study are available in the article and its Supplementary Information. Source data are provided with this paper.

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

## Acknowledgements

This work was financially supported in part by National Key R&D Program of China (2020YFA0406103), NSFC (21725102, 22175165, 22232003, 91961106, 91963108), Strategic Priority Research Program of the CAS (XDPB14), Open Funding Project of National Key Laboratory of Human Factors Engineering (No. SYFD062010K), Youth Innovation Promotion Association CAS (2021451), USTC Research Funds of the Double First-Class Initiative (YD2060002025), Fundamental Research Funds for the Central Universities (WK2340000104, WK2400000004), and China Postdoctoral Science Foundation (2021M703063). The authors thank the support from USTC Center for Micro- and Nanoscale Research and Fabrication. The authors thank BL10B in NSRL for characterizations by synchrotron radiation.

## Author contributions

C.G. and Y.X. supervised the project. F.G. and G.L. designed and performed the experiments. A.C. contributed to electrochemical measurements. Y.H. contributed to transient fluorescence characterization. A.C., H.W., J.P., J.F., H.Z., Y.W., Y.M., C.G. and Y.X. analyzed the data. F.G., G.L., C.G. and Y.X. wrote the paper.

## Competing interests

The authors declare no competing interests.
