## [Peer Review File · Nature Communications]

REVIEWER COMMENTS

Reviewer #1 (Remarks to the Author):

This manuscript describes the construction of a so-called “programmable” artificial photosynthetic cell for CO₂ conversion. This system features the regeneration of NADH with the help of CdTe+. It outcompetes the natural thylakoid that only can regenerate NADPH and ATP. However, this novelty is not very significant and the evidence still has several flaws. It is more like a combination of using abiotic and biotic cofactor regeneration modules in one-pot. The synergy of the two modules and the “programmable” feature are not shown very well. I would suggest a rejection by Nature Communications and the submission to the journal like ACS Catalysis or the similar level ones, after addressing the concerns below.

-paragraphs 1 and 2, The authors claim the system as an artificial photosynthetic cell. However, it lacks lots of characteristics of a cell. At most, it only provides some insights to the development of the artificial photosynthetic cell.

-line 67, Please check more references. Abiotic energy modules should be able to regenerate both NADH and NADPH.

-The preparation method of Tk needs to be double checked. From the images, I doubt that the authors actually obtained the broken chloroplast rather than the Tk, since many Tks are bound with the membrane of chloroplast. Based on the EDS and CLSM, we can see the mapping and merge of the chloroplast membrane. It cannot indicate that the adsorption of CdTe+ on the Tk membrane.

-in Fig S4, the baseline of the adsorption spectra seems to be incorrect.

-in the method, is the light intensity too much? $0.1\text{W}/\text{cm}^2 = 1000\text{W}/\text{m}^2 = 4600\mu\text{moles}/\text{m}^2/\text{s}$, which would lead to large heat generation and may be harmful to Tk.

-in Fig S7, why adding Chl equivalent can increase the amount of NADPH or NADH in the case of CdTe+? Suppose there is no Tk?

-in Fig S12, it should add a control of CdTe+ + [Cp*Rh(bpy)H₂O], so that we can know whether the electron from CdTe+ can be transferred to regenerate NAD or NADP in presence of the mediator.

-in Fig 4i, why no CH₄ can be generated using CdTe+ + MoFe? There is no ATP in the initial reaction?

Reviewer #2 (Remarks to the Author):

In this work, Manolova et al. reported that Tk–CdTe QDs hybrid enabled the regeneration of multiple bioactive cofactors without requiring external supplements. They show this biotic–abiotic hybrid can be coupled with NADPH-dependent PsFDH, NADH-dependent CcFDH, and ATP-dependent MoFe, converting CO₂ into HCOOH and CH₄. Finally, they encapsulate and operate the hybrid energy module in cell-sized droplets. Overall, the authors have developed artificially controllable CO₂ conversion by integration with various reductases. Although the authors’ reported system is quite novel, there are several major issues from the material characterizations to CO₂ photo-conversion that need to be addressed in more detail before it is recommended for publication in Nature Communications. Further, most of the discussions are too light and left as lab reports rather than being deep and informative.

Major comments:

(1) What is the rationale for preparing quantum dots instead of nanoparticles? Especially, the QDs always exhibit a wider band gap which possesses less visible-light absorption; however, stronger visible-light absorption is essential to meaningfully mimic the process of natural photosynthesis. I suggest the authors to explain the rationale for preparing QDs in the introduction. The authors need to address the role of the CdTe particle sizes in the CO₂ photo-conversion performance.

(2) I suggest the authors to provide a higher quality TEM image of CdTe QDs rather than that presented in Supplementary Fig. 1 and also plot the histogram of the particle size.

(3) The title of the y-axis of Supplementary Fig. 4 is not clear. The authors need to plot the absorbance with a proper unit such as “L m⁻¹ g⁻¹” since the UV-visible spectra were measured by dispersing the QDs into buffer D as stated in the Supplementary Information. The current absorption spectra show that the light-harvesting was enhanced at the large wavelength in the range of 700 to 800 nm after incorporation of CdTe QDs (Supplementary Fig. 4). However, the blue curve does not show any absorption for the CdTe QDs. So, this figure should be reexamined.

- (4) Thylakoids are membrane-bound compartments. It is expected that the CdTe QDs are uniformly loaded on this membrane. So, how do the authors define the perimeter for such a biotic material? Do the authors mean there are particular sites for CdTe QDs to be loaded thereon?
- (5) I suggest the authors to quantify the amount of loaded CdTe QDs for the different Tk to CdTe ratios (i.e., 2:1, 1:1, and 1:2). Otherwise, it is very hard for the readers to compare the CdTe-dependent light-driven regeneration of multiple cofactors.
- (6) Why by increasing the added amount of CdTe QDs the loading efficacy of CdTe+ in Tk-CdTe decreases?
- (7) The authors could elaborate a bit more on the volcano profile (or non-linear behavior) of the NADH and ATP regeneration versus the amount of loaded CdTe QDs.
- (8) Did the authors optimize the concentration of chlorophyll? It seems they have measured the cofactors regeneration for 5, 10, and 15 mg ml⁻¹ and conducted the CO₂ photo-conversion test with the chlorophyll concentration of 15 mg ml⁻¹. Why did not they enhance the amount of chlorophyll to obtain better CO₂ photo-conversion performance?
- (9) The results show a monotonic enhancement in the regeneration performance, especially for NADPH and NADH (Fig. 2d-e). Does it mean the generation of charge carriers is the rate-limiting step component (for instance, see Ref. [Appl. Catal. B, 162 (2015) 210-216]) in the semiconducting abiotic?
- (10) In the light intensity experiment in figure 3d (typo error_figure 2d), The linear behavior can be explained as the electron-holes generation is a rate-determining step. What is the relation after increasing light intensity to more than 0.1W/cm²? In addition, thylakoids exhibit a distinct capacity for co-regenerating NADPH and ATP by sunlight. Why in case of ATP, it does not increase with light intensity?
- (11) Does the observed quenching relate to the low amount of CdTe in the Tk-CdTe hybrid? That is, are the concentration of CdTe QDs the same in buffer D during performing the fluorescence experiment?
- (12) Please provide the results of the ICP-AES values for each element in a supplementary table and explain how to calculate the exact ratios between Tk and CdTe.
- (13) How do the authors plot the energy diagram of the hybrid in Fig. 4c? The optical band gap of CdTe QDs should be around ~580 nm based on the fluorescence (Fig.4a and Supplementary Fig. 9) and UV-visible spectroscopies (Supplementary Fig. 4). However, the optical band gap of CdTe QDs is 1.54 (from -1.25 to +0.29 vs NHE) in Fig. 4c.
- (14) In figure 4(d), the transient fluorescence spectra comparison is measured with an emission wavelength of 590 nm. It is not the right way for comparison since there is no fluorescence peak for the TK sample at 590 nm. The evaluation should rely on the two characteristic peaks of each pristine sample and the composite. please include the measurement of CdTe QDs and CdTe+ QDs as well.
- (15) Since the authors did not provide the data related to the concentration of CdTe QDs in buffer D, the reduction in the exciton lifetime obtained from transient fluorescence spectroscopy can be interpreted differently, i.e., not just electron injection to thylakoid. The author should plot the fitted curves inside the transient fluorescence spectra. I further suggest the authors to consider the instrument response function (IRF) for fitting these curves rather than using $i(t) = B1 \cdot \exp(-t/\tau1) + B2 \cdot \exp(-t/\tau2) + A$ (see Ref. [Lab Chip 14 (2014) 4338-4343]).
- (16) What is the contribution of PDADMAC to light absorption? the absorption measurement presented in the supplementary figure 4 reflects the existence of PDADMAC absorption as well as CdTe QDs. Thus, please include the UV-vis absorption measurement of CdTe QDs and CdTe+ QDs and explain the contribution of the Polymer. In addition, does this polymer contribute to the charge transfer mechanism in the hybrid cell to drive the regeneration of the cofactors?
- (17) For most redox enzymes, the cofactors act as electron carriers to transfer electrons between a photocatalyst and enzyme. Here, the electron transfer route in photocatalyst is the photoexcited electrons in CdTe were transferred to thylakoid. Please explain how the cofactors will interact with thylakoid if the TK is shielded by the decoration of CdTe in terms of electron collection and transport.
- (18) The reaction mechanism and the nature of the active site are unclear. For example, what is the oxidation reaction?
- (19) The phrase "programmable" in the title cannot be supported by the reported context and is misleading. I strongly suggest the authors to modify the title.
- (20) The reported numbers for the photochemical CO₂ conversion should be comparable with other reports in the field. Can the authors estimate the number of consumed charge rate per active site for this biotic-abiotic hybrid (for instance, see Ref. [Nat. Commun., 13 (2022) 1256.]?)

Minor comments:

- (1) As stated above, I do not see the motivation/rational of using CdTe QDs in this study. Do other QDs such as CdS, MoS₂, and others work in such a system? Besides, CdTe is known as toxic substance, why the author choose it as compared to other semiconductors?
- (2) The authors should provide high-magnification TEM images displaying the presence of QDs on the thylakoids. The current TEM images (2nd column of Fig. 1a) are not clear. Actually, both this figure and that shown in Figure 2 look more like aggregated CdTe nanoparticles.
- (3) In the supplementary figure 1, What is the size of the QDs? Why the TEM images of the pristine CdTe QDs exhibit an aggregation after treatment with PDADMAC?
- (4) Did the authors perform any background blank tests for CO₂ photo-conversion experiments?

Reviewer #4 (Remarks to the Author):

In the manuscript "Programmable artificial photosynthetic cells with biotic-abiotic hybrid energy modules for customized CO₂ conversion" the authors report artificial photosynthetic cells where thylakoids are interfaced with CdTe quantum dots to obtain an enhanced flux of photogenerated electrons in the photosynthetic electron transfer cycle. This process is performed to drive an enhanced NADPH/NADH and ATP regeneration in the modified thylakoids. The regenerated cofactors are then used to drive CO₂ conversion, and a miniaturized system with artificial photosynthetic cells is also reported. The investigated topic, with the use of biohybrid systems for semi-artificial photosynthesis, is timely and of broad interest. Accordingly, the research topic is suitable for Nature Communications. However, there are various aspects in the manuscript that should be revised, as described in the specific comments below, and the work should be reconsidered after a major revision.

Specific comments:

- 1) Lines 171-200: When discussing the regeneration of NADPH, NADH and ATP by the thylakoid, CdTe QDs, and modified Tk-CdTe QDs, the comparison is performed in view of the results presented in Fig 3 a-c. However, Fig 3 d-f show that there is a clear influence of light intensity on cofactors regeneration. Accordingly, when discussing the results presented in Fig 3 a-c, it should be clearly mentioned that they refer only to a specific light intensity, since for low light intensities pristine Tk performed better than Tk-CdTe for NADPH regeneration (and similarly, CdTe had the same performance for NADH regeneration under low light intensity).
- 2) lines 226-229: When discussing cofactors regeneration over multiple light/dark cycles, the decreased performance of CdTe QDs after 30 min is attributed to QDs aggregation, however, there is no experimental evidence supporting this claim.
- 3) Lines 270-278: When discussing the bioactivity of the regenerated cofactors, it is stated that the non-enzymatic regeneration of the cofactors might results in inactive dimers or isomer. It is then stated that the cofactors regenerated by both pristine thylakoids and CdTe QDs were non-bioactive. While the results for CdTe QDs could be expected, it is not discussed why the enzymatically regenerated cofactors with pristine thylakoids are not bioactive.
- 4) Lines 375-385: when presenting the cofactor regeneration performance in the artificial cells, it would be interesting to also compare NADPH regeneration, and not only NADH, since pristine Tk showed comparable performance to Tk-CdTe 1:1, especially under low light intensity.

Minor comments

- line 323 and 324: error values should be included for the percentages reported.
- in the introduction, the comparison of the approach presented in this work for enhancing the supply of photogenerated electrons to thylakoids with other approaches reported in literature is missing.

Point-by-point response to the reviewers' comments

Reviewer #1:

This manuscript describes the construction of a so-called “programmable” artificial photosynthetic cell for CO₂ conversion. This system features the regeneration of NADH with the help of CdTe⁺. It outcompetes the natural thylakoid that only can regenerate NADPH and ATP. However, this novelty is not very significant and the evidence still has several flaws. It is more like a combination of using abiotic and biotic cofactor regeneration modules in one-pot. The synergy of the two modules and the “programmable” feature are not shown very well. I would suggest a rejection by Nature Communications and the submission to the journal like ACS Catalysis or the similar level ones, after addressing the concerns below.

Author response: We are grateful to the referee for his/her comments and suggestions to help us further improve the quality of our manuscript. We have made all the revisions as suggested by the referees, and would like to emphasize the significance of our work.

The goal of artificial photosynthetic cells for CO₂ conversion is to achieve customizable carbonaceous products toward multifarious demands (Science, 2020, 368, 649-654). However, developing one photoenzymatic platform that can achieve efficient regeneration of multiple cofactors remains a giant challenge. In this work, we designed a novel biotic–abiotic hybrid energy module that offered an efficient capability for multiple cofactors regeneration to power diverse photoenzymatic CO₂ conversion.

Our prepared hybrid energy module (namely, Tk–CdTe) is not just a simple combination of using abiotic and biotic cofactor regeneration modules in one pot. In fact, using abiotic and/or biotic cofactor regeneration modules in one pot cannot provide effective cofactors regeneration without addition of electronic mediators.

Moreover, albeit the abiotic energy modules could power NADH-dependent photoenzymatic reactions, it commonly requires the addition of electronic mediators (Supplementary Table 1). The rational integration of thylakoid with CdTe quantum dots substantially promotes the regeneration of bioactive NADPH, NADH and ATP cofactors without requiring external supplements, remarkably exceeding the individual component (Fig. 3a-c). In particular, CdTe QDs turn thylakoid active for NADH regeneration via electron transfer. This work provides a promising approach to engineering artificial photosynthetic cells by employing the novel biotic–abiotic hybrid energy modules to drive multiple cofactors regeneration, which sets up a platform for photoenzymatic CO₂ conversion toward customizable carbonaceous products. As such, we believe that our work is suitable for publication in Nature Communications.

-paragraphs 1 and 2, The authors claim the system as an artificial photosynthetic cell. However, it lacks lots of characteristics of a cell. At most, it only provides some insights to the development of the artificial photosynthetic cell.

Author response: We thank the referee for his/her insightful comments. Just like the proposed concept in published articles (Science, 2020, 368, 649-654; Nat. Mater. 2018, 17, 89-96; Nat. Chem. 2019, 11, 32-39; Angew. Chem. Int. Ed. 2018, 57, 13382-13392), “artificial cells” are micrometer-sized structures which can mimic characteristics of living cells. To date, artificial cells are primarily defined as two types: membrane-enclosed spherical assemblies and membrane-free droplets that sequester molecular components.

In our constructed system, droplet-based microfluidics allows for monodispersed droplets to mimic cell membrane that provides a boundary between external and internal environment for cellular processes. The inner compartment in our droplets contains energy module (Tk–CdTe) and enzymes for CO₂ conversion. Under light illumination, inner energy module absorbs light energy from external environment to produce NADPH, NADH and ATP that are utilized by reductase to achieve CO₂ reduction, mimicking the unique feature of living photosynthesis cells. As such, our system belongs to membrane-enclosed spherical structure and possesses photosynthesis characteristics that is reasonable to be called “artificial photosynthetic cell”. In addition, we have performed systematic characterizations on both morphology and function (Fig. 5).

-line 67, Please check more references. Abiotic energy modules should be able to regenerate both NADH and NADPH.

Author response: We thank the referee for his/her kind suggestion and thoughtful comment. We have accordingly checked the literature carefully, and totally agree with the referee that abiotic energy modules could regenerate both NADH and NADPH by electrochemical or photochemical reduction methods. However, just as summarized in Supplementary Table 1 in our manuscript, the published work on photoenzymatic CO₂ conversion systems commonly focused on sole NADH or NADPH regeneration system (Chem. Soc. Rev., 2021, 50, 13449-13466; Chem. Soc. Rev., 2022, 51, 6704-6737). To date, developing one energy module that can achieve efficient regeneration of multiple cofactors (NADPH, NADH and ATP) remains a giant challenge and has been rarely reported. In this work, we designed a novel biotic–abiotic hybrid energy module that offered an efficient capability for multiple cofactors (NADPH, NADH and ATP) regeneration to power diverse photoenzymatic CO₂ conversion.

-The preparation method of Tk needs to be double checked. From the images, I doubt that the authors actually obtained the broken chloroplast rather than the Tk, since many

Tks are bound with the membrane of chloroplast. Based on the EDS and CLSM, we can see the mapping and merge of the chloroplast membrane. It cannot indicate that the adsorption of CdTe+ on the Tk membrane.

Author response: We thank the referee for his/her insightful suggestion and comments. The thylakoid (Tk) was prepared by isolating from young spinach according to the reported method (Science 2020, 368, 649-654; Angew. Chem. Int. Ed. 2022, 61, e202111054). The isolating process can be described as four steps: chloroplast isolation from spinach leave, chloroplast purification, thylakoid acquisition, and thylakoid storage. Among them, thylakoid acquisition was performed by breaking the chloroplast in osmotic shock buffer, which obtained intact thylakoid and plasma membrane of broken chloroplast. Further purification was carried out by throwing away broken membrane with repeated washing (described in the methods section, “Preparation of thylakoid”).

According to the comment, Blue native PAGE assay (Nat. Protoc., 2016, 1, 418-428) was carried out to verify the structural and functional integrity of obtained thylakoid. Nine bands including PSI-PSII megacomplex, PSI supercomplex, PSII core dimer, ATPase, PSII core, Cytb6f supercomplex, PSII core monomer less CP43, LHCII trimer and LHCII monomer were found (Supplementary Fig. 4), which is consistent with previous research (Angew. Chem. Int. Ed., 2022, 61, e202111054; J. Exp. Bot., 2007, 58, 3695-3710). These results suggest that intact thylakoid with well-preserved structure and light-harvesting function was obtained.

Supplementary Fig. 4 | Characterizations of thylakoid. BN-PAGE analysis of photosynthetic apparatus isolated from spinach. The thylakoid was solubilized by 2 % n-Dodecyl- β -D-Maltopyranoside (DDM). Thylakoid loaded in lane was 12.5 μ g

(equivalent of Chl). PSI: photosystem I; PSII: photosystem II; Cyt *b6f*: cytochrome *b6f* complex; LHCII: light-harvesting complex of photosystem II; CP43: photosystem II chlorophyll binding subunit.

Furthermore, to examine whether thylakoid has been completely isolated from the membrane of chloroplast, we further investigated the location of plasma membrane during the preparation process of thylakoid. Compared with thylakoid, the thylakoid precursor (i.e., chloroplast) shows apparent plasma membrane structure (Nat. Rev. Mol. Cell Biol., 2004, 5, 198-208), which gives rise to the enrichment of cytoplasmic substance and represents lower zeta potential (Supplementary Fig. 3a-b). Moreover, cell membrane staining (Nature, 2015, 526, 118-121; Nat. Nanotechnol., 2013, 8, 933-938) results reveal that thylakoid precursor rather than thylakoid has obvious out-membrane-bound structure (Supplementary Fig. 3c), while the isolated thylakoid shows individual spherical stack membrane (Supplementary Fig. 3d). This confirms that thylakoid has been completely isolated from the membrane of chloroplast.

Supplementary Fig. 3 | TEM image of isolated thylakoid. (a, b) Ultrastructure of (a) thylakoid precursor (i.e., chloroplast, abbr., Chlo) and (b) thylakoid (abbr., Tk). The white arrows indicate intact plasma membrane. The inset in (b) is the comparison of zeta potential between Chlo and Tk. The error bars represent the standard deviations of data from triplicate measurements. $P < 0.05$ (analyzed by two-sided Student's t-test) indicates the significant difference between Chlo and Tk. (c, d) TEM images of phosphotungstic acid stained (c) Chlo and (d) Tk. The white arrows indicate (c) broken plasma membrane of Chlo and (d) individual spherical stack membrane of thylakoid.

The adsorption of CdTe QDs on thylakoid membrane was confirmed by combined results of TEM, EDS and UV–Vis absorption spectra. Upon forming the hybrid structure of Tk–CdTe, EDS mapping revealed that the sample contains the elements of magnesium from thylakoid as well as tellurium and cadmium of CdTe QDs (Fig. 2b). Moreover, the formation of Tk–CdTe hybrid structure through electrostatic interaction ensured homogeneous deposition of CdTe QDs on thylakoid (Supplementary Fig. 7). From the UV–Vis absorption spectra (Supplementary Fig. 8), Tk–CdTe possesses a broad range of light absorption that well combines those of thylakoid and CdTe QDs.

The supplemented results (Supplementary Fig. 3, 7) and discussion have been provided in the revised manuscript.

Supplementary Fig. 7 | HRTEM image of CdTe⁺ located on the thylakoid. The circles outlined by red dashed lines indicate the magnified region.

-in Fig S4, the baseline of the adsorption spectra seems to be incorrect.

Author response: We thank the referee for his/her thoughtful comment. In the original manuscript, Supplementary Fig. 4 shows the normalized data. The raw (a) and normalized (b) UV–Vis absorption spectra are shown as follows for comparison. Note that the adsorption spectra of buffer D is the baseline, which shows no light absorption in our detection range.

-in the method, is the light intensity too much? $0.1\text{W/cm}^2=1000\text{W/m}^2=4600\mu\text{moles/m}^2/\text{s}$, which would lead to large heat generation and may be harmful to Tk.

Author response: We thank the referee for his/her thoughtful comment. In fact, we have used the LED light source with AM 1.5 optical filter for illumination, which leads to limited heat generation. According to the comment, we have monitored the temperature of the reaction system during 90-min light illumination by using thermal imaging camera (the newly added Supplementary Fig. 11). The results indicate that the temperature of the reaction system under the light intensity of 0.1 W/cm^2 was as low as the ambient temperature and it has no obvious change throughout the reaction. As such, there is no significant damage to Tk and no significant photothermal effect on NADPH regeneration by thylakoid under the light intensity of 0.1 W/cm^2 . This conclusion is also confirmed by the results in stability test (Fig. 3g-i and Supplementary Fig. 14, 15).

Supplementary Fig. 11 | Temperature monitoring of the reaction system along with light illumination time. (a) The temperature monitored by using thermal imaging camera at various light illumination time under a light intensity of 0.1 W/cm^2 . (b) Data statistics of monitored temperature with various illumination time.

Supplementary Fig. 14 | Photostability of Tk-CdTe after 30 min light illumination. (a–c) Normalized UV–Vis absorption spectra of (a) CdTe⁺, (b) Tk and (c) Tk-CdTe 1:1 in NADPH regeneration buffer before and after illumination. (d–f) Photographs of (d) CdTe⁺, (e) Tk and (f) Tk-CdTe 1:1 dispersion solution before and after illumination under the light intensity of 0.1 W/cm².

Supplementary Fig. 15 | TEM and SEM images of CdTe⁺, Tk and Tk-CdTe 1:1. (a) Before and (b) after 30 min illumination under the light intensity of 0.1 W/cm².

-in Fig S7, why adding Chl equivalent can increase the amount of NADPH or NADH in the case of CdTe⁺? Suppose there is no Tk?

Author response: We thank the referee for his/her insightful comment. The Chl

concentration was commonly used to normalize the amounts of added thylakoid (Science, 2020, 368, 649-654; Angew. Chem. Int. Ed., 2022, 61, e202111054). Similarly, in our work, Chl concentration was used to normalize the amounts of added thylakoid in our Tk-CdTe. In Fig. S7, in the case of CdTe⁺, no Chl equivalent was added. We have noticed that the legend in original Fig. S7 is misleading. According to the comment, we have modified all the related figures in the revised manuscript.

*-in Fig S12, it should add a control of CdTe⁺ + [Cp*Rh(bpy)H₂O], so that we can know whether the electron from CdTe⁺ can be transferred to regenerate NAD or NADP in presence of the mediator.*

Author response: We thank the referee for his/her thoughtful suggestion. According to the suggestion, we have added a control of CdTe⁺ + [Cp*Rh(bpy)H₂O] (the newly added Supplementary Fig. 27). To suppress undesirable side reactions in non-enzymatic cofactors regeneration, the electron mediator of pentamethylcyclopentadienyl rhodium bipyridine ([Cp*Rh(bpy)H₂O]²⁺, denoted as [M]) has been frequently used as hydride transfer species to exclusively form bioactive cofactors. Indeed, CdTe QDs alone could produce bioactive cofactor with [M] as a supplementary, suggesting that the electrons from CdTe⁺ can be transferred to regenerate NAD or NADP in presence of the mediator. However, it requires a high amount of CdTe QDs (51.84 µg/mL) and [M] (0.4 mM) to produce equal amount of NADH compared with Tk–CdTe.

Supplementary Fig. 27 | Electronic mediator $[\text{Cp}^*\text{Rh}(\text{bpy})\text{H}_2\text{O}]^{2+}$ (denoted as $[\text{M}]$) showing a promoted effect on NADH regeneration by CdTe^+ . (a) The ^1H NMR spectrum of $[\text{Cp}^*\text{Rh}(\text{bpy})\text{Cl}]\text{Cl}$ in CDCl_3 . (b-f) The ^1H NMR spectrum of regenerated NADH with NAD^+ as a substrate in the presence of CdTe^+ and $[\text{M}]$ after 30 min illumination. Standard spectra of (b) NAD^+ and (c) NADH . Spectra in the presence of (d) CdTe^+ at $12.96 \mu\text{g/mL}$ and 0.2 mM $[\text{M}]$, (e) CdTe^+ at $51.84 \mu\text{g/mL}$ and 0.4 mM $[\text{M}]$ and (f) CdTe^+ at $51.84 \mu\text{g/mL}$ and without $[\text{M}]$ added.

-in Fig 4i, why no CH_4 can be generated using $\text{CdTe}^+ + \text{MoFe}$? There is no ATP in the initial reaction?

Author response: We thank the referee for his/her thoughtful questions. Remodeled nitrogenase (MoFe) is an ATP-requiring enzyme having the capacity to reduce carbon

dioxide to methane (Proc. Natl. Acad. Sci. USA., 2012, 109, 19644-19648; Proc. Natl. Acad. Sci. USA., 2016, 113, 10163-10167). As CdTe⁺ alone cannot produce ATP under light illumination due to the lack of ATP synthase, there is no ATP in the initial reaction and thus no CH₄ can be generated by using CdTe⁺ + MoFe.

Reviewer #2:

In this work, Manolova et al. reported that Tk–CdTe QDs hybrid enabled the regeneration of multiple bioactive cofactors without requiring external supplements. They show this biotic–abiotic hybrid can be coupled with NADPH-dependent PsFDH, NADH-dependent CcFDH, and ATP-dependent MoFe, converting CO₂ into HCOOH and CH₄. Finally, they encapsulate and operate the hybrid energy module in cell-sized droplets. Overall, the authors have developed artificially controllable CO₂ conversion by integration with various reductases. Although the authors' reported system is quite novel, there are several major issues from the material characterizations to CO₂ photo-conversion that need to be addressed in more detail before it is recommended for publication in Nature Communications. Further, most of the discussions are too light and left as lab reports rather than being deep and informative.

Author response: We really appreciate the referee's highly positive evaluation of our work, and are grateful to the referee for his/her comments and suggestions to help us further improve the quality of our manuscript. We have made all the revisions as suggested by the referees.

Major comments:

(1) What is the rationale for preparing quantum dots instead of nanoparticles? Especially, the QDs always exhibit a wider band gap which possesses less visible-light absorption; however, stronger visible-light absorption is essential to meaningfully mimic the process of natural photosynthesis. I suggest the authors to explain the rationale for preparing QDs in the introduction. The authors need to address the role of the CdTe particle sizes in the CO₂ photo-conversion performance.

Author response: We thank the referee for his/her thoughtful comments and suggestions. Thylakoid as a biotic energy module offers unparalleled opportunities for multiple cofactors regeneration. However, thylakoid generally suffers from low utilization of light, which not only limits the efficiency of the whole photosynthetic process, but also poses a challenge for increasing the supply of photogenerated electrons (Nat. Nanotechnol., 2018, 13, 890-899; Nat. Mater., 2014, 13, 400-408). Thus, we envisaged that efficient regeneration of cofactors can be achieved by rationally combining thylakoid and light-harvesting inorganic nanomaterials to increase the supply of photogenerated electrons, which could eventually breaks the existing bottlenecks.

Among various light-harvesting inorganic nanomaterials, the light absorption and electronic band structure of quantum dots are more readily tuned by size control. Quantum dots also have more negative conduction band potential compared to the corresponding nanoparticles, which could facilitate the photogenerated electrons transfer to thylakoid (Science, 2021, 373, eaaz8541; Science, 2016, 352, 448-450). For this reason, we chose quantum dots for integration with thylakoid. According to the

suggestion, we have explained the rationale for preparing QDs in the introduction and address the role of the CdTe particle sizes in the CO₂ photo-conversion performance in the revised manuscript.

(2) I suggest the authors to provide a higher quality TEM image of CdTe QDs rather than that presented in Supplementary Fig. 1 and also plot the histogram of the particle size.

Author response: We thank the referee for his/her insightful suggestion. According to the suggestion, we have provided a higher-quality TEM image of CdTe QDs and plotted the histogram of the particle size in the revised manuscript (the updated Supplementary Fig. 1).

Supplementary Fig. 1 | Characterizations of positive charge CdTe QDs (CdTe⁺). (a) Schematic illustration for the preparation of CdTe QDs. (b) Photograph of prepared CdTe QDs dispersion. (c) TEM image of CdTe QDs. (d) Statistical size distribution of the as-synthesized CdTe QDs. (e) HRTEM image of CdTe QDs. (f) Zeta-potential profiles of CdTe QDs. (g) TEM image of CdTe⁺. (h) Statistical size distribution of CdTe⁺. (i) HRTEM image of CdTe⁺. (j) Zeta-potential profiles of CdTe⁺. The error bars represent the standard deviations of data from triplicate measurements.

(3) The title of the y-axis of Supplementary Fig. 4 is not clear. The authors need to plot the absorbance with a proper unit such as “L m⁻¹ g⁻¹” since the UV-visible spectra were measured by dispersing the QDs into buffer D as stated in the Supplementary Information. The current absorption spectra show that the light-harvesting was enhanced at the large wavelength in the range of 700 to 800 nm after incorporation of CdTe QDs (Supplementary Fig. 4). However, the blue curve does not show any absorption for the CdTe QDs. So, this figure should be reexamined.

Author response: We thank the referee for his/her constructive suggestion. In the original manuscript, we have normalized the y-axis absorbance data in Supplementary Fig. 4. The raw (a) and normalized (b) UV–Vis absorption spectra are shown as follows for comparison.

Actually, the individual CdTe QDs do not show any absorption at the large wavelength in the range of 700 to 800 nm. After the incorporation of CdTe QDs onto thylakoid, the Tk–CdTe shows enhanced light absorption in the range of 700 to 800 nm (the updated Supplementary Fig. 8a), although there was no contributed absorption of CdTe QDs in that range. Similar phenomenon was observed when pure PDADMAC was adsorbed on thylakoid (the newly added Supplementary Fig. 8b), indicating that the slightly promoted light-harvesting capability is most likely ascribed to the introduced PDADMAC. However, the introduced PDADMAC has no significant contribution to cofactors regeneration (the newly added Supplementary Fig. 12).

Supplementary Fig. 8 | UV-Vis absorption spectra. (a) Normalized UV-Vis absorption spectra of Tk-CdTe in buffer D. The spectra of individual component including CdTe QDs, CdTe⁺, Tk and buffer D were obtained as the control. (b) UV-Vis absorption spectra of Tk-PDADMAC in buffer D. The spectra of individual component including PDADMAC, Tk and buffer D were obtained as the control. The buffer D includes 700 mM sorbitol, 10 mM HEPES-KOH pH 7.6, 10 mM MgCl₂, and 10 mM sodium L-ascorbate.

(4) *Thylakoids are membrane-bound compartments. It is expected that the CdTe QDs are uniformly loaded on this membrane. So, how do the authors define the perimeter for such a biotic material? Do the authors mean there are particular sites for CdTe QDs to be loaded thereon?*

Author response: We thank the referee for his/her thoughtful comments. Thylakoid consists of the membrane and the enclosed region called the lumen. A stack of thylakoids forms a group of coin-like structures called a granum. In this work, our prepared thylakoid exhibited a spherical stacked structure that was linked by stroma lamellae, without the chloroplast plasma membrane. Although it is difficult to define the inner perimeter of individual thylakoid because of its stacked structure, the globule out-layer can be distinguished easily. Moreover, the perimeter of stacked thylakoid as an integral whole can be observed by SEM, TEM and CLSM clearly (Fig. 2, the newly added Supplementary Fig. 3b and Supplementary Fig. 3d).

The thylakoid membrane mainly contains lipids and photosynthetic protein complexes, and exhibits negative surface charge mainly attributed to the exposed charged amino acids of the photosynthetic protein (Cell Biochem. Biophys., 2020, 78:401-414; Biochim. Biophys. Acta, 2008, 1778, 2823-2833). The presence of negative charges on thylakoid membrane enables the adsorption of positively charged CdTe⁺ QDs through electrostatic force, which is confirmed by SEM and TEM images (Fig. 2a and Supplementary Fig. 7). The particular sites for loading CdTe⁺ QDs on thylakoid membrane could be the negative charged carboxyl in lipids or photosynthetic protein.

Currently it is hard to exactly determine the particular sites due to the complex chemical composition of thylakoid membrane, thus requiring further exploration in future.

Supplementary Fig. 3 | TEM image of isolated thylakoid. (a, b) Ultrastructure of (a) thylakoid precursor (i.e., chloroplast, abbr., Chlo) and (b) thylakoid (abbr., Tk). The white arrows indicate intact plasma membrane. The inset in (b) is the comparison of zeta potential between Chlo and Tk. The error bars represent the standard deviations of data from triplicate measurements. $P < 0.05$ (analyzed by two-sided Student's t-test) indicates the significant difference between Chlo and Tk. (c, d) TEM images of phosphotungstic acid stained (c) Chlo and (d) Tk. The white arrows indicate (c) broken plasma membrane of Chlo and (d) individual spherical stack membrane of thylakoid.

(5) I suggest the authors to quantify the amount of loaded CdTe QDs for the different Tk to CdTe ratios (i.e., 2:1, 1:1, and 1:2). Otherwise, it is very hard for the readers to compare the CdTe-dependent light-driven regeneration of multiple cofactors.

Author response: We thank the referee for his/her thoughtful suggestion. According to the suggestion, we have quantified the amount of loaded CdTe QDs in Tk–CdTe with different Tk to CdTe ratios by ICP–AES and calculated the loading efficacy in the revised manuscript (Supplementary Fig. 6 and the newly added Supplementary Table 2).

Supplementary Table 2 | A summary on the element contents and loading efficacy in Tk–CdTe determined by ICP–AES.

Sample	Cd ($\mu\text{g/mL}$)	Te ($\mu\text{g/mL}$)	Cd/Te	CdTe ($\mu\text{g/mL}$)	Added ($\mu\text{g/mL}$)	Loading (%)
Tk–CdTe						
2:1						
1	231.15	43.86	5.27	275.01	300	91.67
2	219.63	41.37	5.31	261	300	87
3	248.91	47.1	5.28	296.01	300	98.67
4	232.59	44.4	5.24	276.99	300	92.33
Mean \pm s.d.	233.07 \pm 12.05	44.18 \pm 2.35	5.28 \pm 0.03	277.25 \pm 14.39	300	92.42 \pm 4.8
Tk–CdTe						
1:1						
1	352.83	67.14	5.26	419.97	500	83.99
2	373.62	71.37	5.23	444.99	500	89
3	357.09	67.92	5.26	425.01	500	85
4	367.71	70.26	5.23	437.97	500	87.59
Mean \pm s.d.	362.82 \pm 9.54	69.17 \pm 1.98	5.25 \pm 0.01	431.99 \pm 11.51	500	86.40 \pm 2.3
Tk–CdTe						
1:2						
1	467.91	89.13	5.25	557.04	750	74.27
2	490.89	93.18	5.27	584.07	750	77.87
3	500.52	95.43	5.24	596.95	750	79.47
4	476.76	91.2	5.23	567.96	750	75.73
Mean \pm s.d.	484.02 \pm 14.51	92.24 \pm 2.70	5.25 \pm 0.02	576.26 \pm 17.19	750	76.83 \pm 2.29

(6) Why by increasing the added amount of CdTe QDs the loading efficacy of CdTe⁺ in Tk-CdTe decreases?

Author response: We thank the referee for his/her thoughtful question. The loading efficacy is calculated as follows: Loading efficacy = (Loading amount/added amount) \times 100%. Note that the amount of Tk is constant. When increasing the added amount of CdTe⁺ from 300 $\mu\text{g/mL}$ to 750 $\mu\text{g/mL}$, the loading amount of CdTe⁺ increased from 277.25 \pm 14.39 $\mu\text{g/mL}$ to 576.26 \pm 17.19 $\mu\text{g/mL}$ (Supplementary Table 2). This disproportionate increase is most likely due to the near saturated absorption of CdTe⁺ on thylakoid membrane. For this reason, the loading efficacy of CdTe⁺ in Tk–CdTe decreases.

(7) The authors could elaborate a bit more on the volcano profile (or non-linear behavior) of the NADH and ATP regeneration versus the amount of loaded CdTe QDs.

Author response: We thank the referee for his/her kind suggestion. When increasing the loading amount of CdTe QDs, the performance of Tk–CdTe in both NADH and ATP regeneration exhibited a volcano profile. This is most likely due to the limited amount of electron transfer medium in thylakoid or the hindered light absorption by over decoration of CdTe QDs. According to the suggestion, we have included the related discussion in the revised manuscript.

(8) *Did the authors optimize the concentration of chlorophyll? It seems they have measured the cofactors regeneration for 5, 10, and 15 mg ml⁻¹ and conducted the CO₂ photo-conversion test with the chlorophyll concentration of 15 mg ml⁻¹. Why did not they enhance the amount of chlorophyll to obtain better CO₂ photo-conversion performance?*

Author response: We thank the referee for his/her insightful question. In the original manuscript, although we conducted the cofactors regeneration test with the Chl concentration of 15 µg/mL, we have conducted the photoenzymatic CO₂ conversion inside artificial photosynthetic cells with the Chl concentration of 50 or 80 µg/mL (Supplementary Table 11) and obtained better cofactors regeneration performance (Fig. 5d and Supplementary Fig. 48, 49). According to the comment, we have systemically optimized the concentration of chlorophyll on cofactors regeneration (the updated Fig. 3a–c). The cofactors regeneration yields over Tk and Tk–CdTe are both positively correlated with the concentration of Chl equivalent in the range of 5 to 120 µg/mL, while the yields have no sharp growth when the concentration of Chl equivalent increased to 120 µg/mL. This slow growth is most likely ascribed to the limited NADP⁺ concentration and illumination area. The related data and discussions have been supplemented in the revised manuscript.

(9) *The results show a monotonic enhancement in the regeneration performance, especially for NADPH and NADH (Fig. 2d-e). Does it mean the generation of charge carriers is the rate-limiting step component (for instance, see Ref. [Appl. Catal. B, 162 (2015) 210-216]) in the semiconducting abiotic?*

Author response: We thank the referee for his/her insightful comment and question. According to the suggestion, we have examined the cofactor regeneration with the light intensity at 0.01–0.3 W/cm² (the updated Fig. 3d–f). We found that the regeneration yields of NADPH and NADH by Tk–CdTe show a nearly monotonic enhancement with increasing light intensity at 0.01–0.2 W/cm². According to the literature provided by the referee, such a monotonic enhancement suggests that charge carriers generation may be the rate-limiting step. We have now included the discussion and reference (the newly added Ref. 19) in the revised manuscript.

(10) In the light intensity experiment in figure 3d (typo error_figure 2d), The linear behavior can be explained as the electron-holes generation is a rate-determining step. What is the relation after increasing light intensity to more than 0.1W/cm²? In addition, thylakoids exhibit a distinct capacity for co-regenerating NADPH and ATP by sunlight. Why in case of ATP, it does not increase with light intensity?

Author response: We thank the referee for his/her insightful comment and question. According to the suggestion, we have examined the cofactor regeneration after increasing light intensity to more than 0.1 W/cm² (the updated Fig. 3d–f). We found that the regeneration yields of cofactors by Tk–CdTe were promoted by increasing light intensity, although pristine thylakoid exhibited the similar tendency of light-dependent cofactors regeneration to the literature (Angew. Chem. Int. Ed., 2022, 61, e202111054). The enhancement of NADPH regeneration was thoroughly quantified as a function of the light intensity. The production yields of NADPH were elevated from 11.95 ± 1.12 to 98.34 ± 1.85 μM by increasing light intensity at 0.01–0.3 W/cm². Similarly, the NADH or ATP regeneration was also enhanced with increasing light intensity (Fig. 3e and 3f). However, when the light intensity increased to 0.2 W/cm², there is no obvious enhancement on cofactor regeneration, which is most likely due to the self-protection mechanism (cyclic electron flow) in photosynthetic system of thylakoid (Proc. Natl. Acad. Sci. U.S.A., 2011, 108, 13317-13322).

Previous reports indicated that the regeneration yields of cofactors in photoenzymatic processes were limited by light saturation (Nat. Nanotechnol., 2018, 13, 890-899; Angew. Chem. Int. Ed., 2022, 61, e202111054). Consistently, the cofactor regeneration capability of thylakoid in our work is not significantly influenced by the increasing light intensity. Notably, as the light intensity varied, the sample of CdTe QDs showed the consistent tendency with Tk–CdTe despite its inferior performance (Fig. 3d and 3e). These results demonstrate that the coupling of thylakoid with CdTe QDs endowed the hybrid energy modules with enhanced cofactors regeneration performance.

Fig. 3d–f | Light-driven regeneration of NADPH, NADH and ATP cofactors by thylakoid–CdTe energy modules. (d) NADPH, (e) NADH and (f) ATP regeneration under different light intensities by CdTe⁺, Tk or Tk–CdTe 1:1 with 10 μg/mL Chl equivalent. The amounts of added CdTe⁺ was 8.64 μg/mL. The error bars represent the standard deviations of data from triplicate measurements.

(11) Does the observed quenching relate to the low amount of CdTe in the Tk–CdTe hybrid? That is, are the concentration of CdTe QDs the same in buffer D during performing the fluorescence experiment?

Author response: We thank the referee for his/her thoughtful questions. We have quantified the amount of loaded CdTe QDs in Tk–CdTe with different Tk to CdTe ratios by ICP–AES and calculated the loading efficacy in the revised manuscript (Supplementary Fig. 6 and the newly added Supplementary Table 2). During performing the fluorescence experiment, the concentration of CdTe QDs was indeed kept the same in buffer D to ensure a fair comparison.

Supplementary Table 2 | A summary on the element contents and loading efficacy in Tk–CdTe determined by ICP–AES.

Sample	Cd ($\mu\text{g/mL}$)	Te ($\mu\text{g/mL}$)	Cd/Te	CdTe ($\mu\text{g/mL}$)	Added ($\mu\text{g/mL}$)	Loading (%)
Tk–CdTe						
2:1						
1	231.15	43.86	5.27	275.01	300	91.67
2	219.63	41.37	5.31	261	300	87
3	248.91	47.1	5.28	296.01	300	98.67
4	232.59	44.4	5.24	276.99	300	92.33
Mean \pm s.d.	233.07 \pm 12.05	44.18 \pm 2.35	5.28 \pm 0.03	277.25 \pm 14.39	300	92.42 \pm 4.8
Tk–CdTe						
1:1						
1	352.83	67.14	5.26	419.97	500	83.99
2	373.62	71.37	5.23	444.99	500	89
3	357.09	67.92	5.26	425.01	500	85
4	367.71	70.26	5.23	437.97	500	87.59
Mean \pm s.d.	362.82 \pm 9.54	69.17 \pm 1.98	5.25 \pm 0.01	431.99 \pm 11.51	500	86.40 \pm 2.3
Tk–CdTe						
1:2						
1	467.91	89.13	5.25	557.04	750	74.27
2	490.89	93.18	5.27	584.07	750	77.87
3	500.52	95.43	5.24	596.95	750	79.47
4	476.76	91.2	5.23	567.96	750	75.73
Mean \pm s.d.	484.02 \pm 14.51	92.24 \pm 2.70	5.25 \pm 0.02	576.26 \pm 17.19	750	76.83 \pm 2.29

(12) Please provide the results of the ICP-AES values for each element in a

supplementary table and explain how to calculate the exact ratios between Tk and CdTe.

Author response: We thank the referee for his/her thoughtful suggestion. According to the suggestion, we have quantified the amount of loaded CdTe QDs in Tk–CdTe with different Tk to CdTe ratios by ICP–AES and calculated the loading efficacy in the revised manuscript (Supplementary Fig. 6 and the newly added Supplementary Table 2). The amount of Tk was indicated by the amount of Chl, which was determined according to the previously reported method (Photosyn. Res., 2002, 73, 149-156), while the exact amount of CdTe was determined by ICP–AES. As such, we can calculate the exact ratios between Tk and CdTe.

(13) How do the authors plot the energy diagram of the hybrid in Fig. 4c? The optical band gap of CdTe QDs should be around ~580 nm based on the fluorescence (Fig.4a and Supplementary Fig. 9) and UV-visible spectroscopies (Supplementary Fig. 4). However, the optical band gap of CdTe QDs is 1.54 (from -1.25 to +0.29 vs NHE) in Fig. 4c.

Author response: We thank the referee for his/her insightful comment and question. In our original manuscript, we plotted the energy diagram according to previous reports (ACS Nano, 2009, 3, 1467-1476; J. Phys. Chem. C, 2016, 120, 650-658; J. Am. Chem. Soc., 1977, 99, 2839-2848) that were based on the values of bulk CdTe. As for CdTe QDs, the quantum confinement increases the effective bandgap, leading to a blue shift of the absorption and emission spectra (Science, 2021, 373, eaaz8541).

In this work, the optical band gap of prepared CdTe QDs (3.8 nm) is determined to be 1.86 V (Supplementary Fig. 23a, b), which is larger than bulk. In the revised manuscript, electrochemical measurement has been carried out to determine the conduction band (CB) and valence band (VB) energy levels of prepared CdTe QDs. The calculated CB and VB value are -1.5 and 0.36 (V, vs NHE), respectively, which is consistent with previous research (ACS Nano, 2009, 3, 1467-1476; J. Phys. Chem. C, 2016, 120, 650-658; J. Am. Chem. Soc., 1997, 99, 2839-2848). In addition, the CB potential of PSII is reported to be -0.58 V (vs. NHE) (Proc. Natl. Acad. Sci. U.S.A., 2016, 113, 620-625; Small, 2018, 14, 180010), and as such, the photogenerated electrons can favorably transfer from CdTe QDs to Tk. According to the comment, we have revised the energy diagram and included the related data in the revised manuscript.

Supplementary Fig. 23 | Calculation of energy band diagrams of CdTe QDs. (a, b) UV-Vis diffuse reflectance spectra and (inset) calculated band gap of (a) CdTe QDs and (b) CdTe⁺. Optical band gap was determined with a Tauc plot. Linear sweep voltammetry (LSV) curves of (c) CdS QDs and (d) CdTe⁺.

Fig. 4 | (i) Schematic illustration of the redox potentials of CdTe⁺ and PS II (top), and a possible mechanism for electron transfer from excited CdTe⁺ to PS II (down). Note that band gap and positions of CdTe⁺ was calculated with UV-Vis diffuse reflectance spectra and linear sweep voltammetry (LSV). PSII is depicted based on previous reports.

(14) In figure 4(d), the transient fluorescence spectra comparison is measured with an emission wavelength of 590 nm. It is not the right way for comparison since there is no fluorescence peak for the TK sample at 590 nm. The evaluation should rely on the two characteristic peaks of each pristine sample and the composite. please include the measurement of CdTe QDs and CdTe⁺ QDs as well.

Author response: We thank the referee for his/her thoughtful comment and suggestion. According to the suggestion, we have re-performed the transient fluorescence spectra comparison relying on the two characteristic peaks of each pristine sample and the composite, and included the measurement of CdTe QDs and CdTe⁺ QDs as well (the updated Fig. 4c, d). The related discussions and detailed fitting parameters for the fluorescence decay curves (the newly added Supplementary Table 3, 4) have been provided in the revised manuscript.

Fig. 4 | Light-driven CO₂ conversion by integrating reductases with cofactors regeneration using Tk–CdTe. (c, d) Transient fluorescence spectra of (c) CdTe QDs, CdTe⁺ and Tk–CdTe 1:1 with an emission wavelength of 590 nm and (d) Tk and Tk–CdTe 1:1 with an emission wavelength of 685 nm. “Prompt” represents the instrument response function.

Supplementary Table 3 | Fitting parameters for the fluorescence decay curves of CdTe in Tk–CdTe.

Sample	T1 (B1)	T2 (B2)	T3 (B3)	τ_1 (ns)	τ_2 (ns)	τ_3 (ns)	τ (ns)	χ^2
CdTe	14.50 (47.74%)	58.16 (28.63%)	3.30 (23.62%)	6.92	16.65	0.78	24.35	1.19
CdTe ⁺	14.85 (35.11%)	41.82 (53.68%)	2.54 (11.21%)	5.21	22.45	0.29	27.95	1.15
Tk–CdTe	5.34	32.36	0.24	1.70	5.78	0.12	7.60	0.99

(31.91%) (17.85%) (50.24%)

Supplementary Table 4 | Fitting parameters for the fluorescence decay curves of Tk in Tk–CdTe.

Sample	T1 (B1)	T2 (B2)	T3 (B3)	τ 1 (ns)	τ 2 (ns)	τ 3 (ns)	τ (ns)	χ^2
Tk	1.30 (34.82%)	0.22 (62.24%)	5.64 (2.94%)	0.45	0.14	0.17	0.76	1.01
Tk–CdTe	1.35 (32.72%)	0.24 (61.45%)	8.41 (5.83%)	0.44	0.15	0.49	1.08	1.00

(15) Since the authors did not provide the data related to the concentration of CdTe QDs in buffer D, the reduction in the exciton lifetime obtained from transient fluorescence spectroscopy can be interpreted differently, i.e., not just electron injection to thylakoid. The author should plot the fitted curves inside the transient fluorescence spectra. I further suggest the authors to consider the instrument response function (IRF) for fitting these curves rather than using $i(t) = B1 * \exp(-t/\tau1) + B2 * \exp(-t/\tau2) + A$ (see Ref. [Lab Chip 14 (2014) 4338-4343]).

Author response: We thank the referee for his/her thoughtful comment and suggestion. According to the suggestion, we have quantified the amount of loaded CdTe QDs in Tk–CdTe with different Tk to CdTe ratios by ICP–AES in the revised manuscript (the newly added Supplementary Table 2). During performing the fluorescence experiment, the concentration of CdTe QDs was indeed kept the same in buffer D to ensure a fair comparison.

Additionally, we have used the instrument response function (IRF) for fitting the fluorescence decay curves. Based on the information conveyed in Horiba user guide (<https://www.horiba.com/fileadmin/uploads/Scientific/Downloads/UserArea/Fluorescence/Manuals/DAS67-Manual.pdf>) and previously reported methods (Nat. Commun., 2021, 12, 1364; Angew. Chem. Int. Ed., 2021, 60, 16970-16973; J. Phys. Chem. C, 2016, 120, 650-658), we have re-performed the transient fluorescence spectra comparison relying on the characteristic peaks of each pristine sample and re-fitted the fluorescence decay curves by considering the instrument response function (IRF) (i.e., prompt). The quality of the fitting was assessed by the χ^2 values (which should be near 1.0 and < 1.2 , Supplementary Table 3, 4), showing the good fitting of our results. The updated results are also consistent with previous research (Small, 2012, 8, 17, 2652-2658; J. Phys. Chem. C, 2016, 120, 650-658; Plant Cell Rep., 2017, 36, 327-341; Methods Appl. Fluoresc., 2020, 8, 024007).

(16) What is the contribution of PDADMAC to light absorption? the absorption

measurement presented in the supplementary figure 4 reflects the existence of PDADMAC absorption as well as CdTe QDs. Thus, please include the UV-vis absorption measurement of CdTe QDs and CdTe⁺ QDs and explain the contribution of the Polymer. In addition, does this polymer contribute to the charge transfer mechanism in the hybrid cell to drive the regeneration of the cofactors?

Author response: We thank the referee for his/her insightful questions and constructive suggestion. According to the suggestion, we have included the UV-Vis absorption measurement of CdTe QDs, CdTe⁺ QDs and PDADMAC to explain the contribution of the polymer.

PDADMAC alone does not show any absorption at the wavelength in the range of 350 to 800 nm. Interestingly, CdTe⁺ indicated slightly reduced absorption at the wavelength of ~540 nm after the PDADMAC modification (Supplementary Fig. 8a). This is most likely due to the reduced absolute value of zeta potential for CdTe QDs after surface modification of PDADMAC (from -50.1 mV to 30.1 mV, Supplementary Fig. 1), which leads to weaker repulsive force, shortened space distance of the colloid and thus the changed light absorption peak.

After the incorporation of CdTe QDs on thylakoid, the Tk-CdTe shows enhanced light absorption in the range of 700 to 800 nm (the updated Supplementary Fig. 8a), although there was no contributed absorption of CdTe QDs in that range. Similar phenomenon was observed when pure PDADMAC was absorbed on thylakoid (the newly added Supplementary Fig. 8b), indicating that the slightly promoted light-harvesting capability should be ascribed to interaction between introduced PDADMAC and light-harvesting protein. The underlying mechanism needs be further detailedly explored in future. However, the introduced PDADMAC has no significant contribution to cofactors regeneration (the newly added Supplementary Fig. 12), suggesting that this polymer has no significant contribution to the charge transfer.

Supplementary Fig. 8 | UV-Vis absorption spectra. (a) Normalized UV-Vis absorption spectra of Tk-CdTe in buffer D. The spectra of individual component

including CdTe QDs, CdTe⁺, Tk and buffer D were obtained as the control. (b) UV–Vis absorption spectra of Tk–PDADMAC in buffer D. The spectra of individual component including PDADMAC, Tk and buffer D were obtained as the control. The buffer D includes 700 mM sorbitol, 10 mM HEPES-KOH pH 7.6, 10 mM MgCl₂, and 10 mM sodium L-ascorbate.

Supplementary Fig. 12 | The contribution of PDADMAC to cofactors regeneration.

(a) Fluorescence spectra of Tk–PDADMAC in buffer D. The spectra of PDADMAC, Tk and buffer D were collected as the control. (b) NADPH and NADH regeneration by Tk–PDADMAC or Tk with 10 μg/mL Chl equivalent. The error bars represent the standard deviations of data from triplicate measurements. $P > 0.05$ (analyzed by two-sided Student’s t-test) indicates no significant difference between Tk–PDADMAC and Tk for cofactors regeneration.

(17) For most redox enzymes, the cofactors act as electron carriers to transfer electrons between a photocatalyst and enzyme. Here, the electron transfer route in photocatalyst is the photoexcited electrons in CdTe were transferred to thylakoid. Please explain how the cofactors will interact with thylakoid if the TK is shielded by the decoration of CdTe in terms of electron collection and transport.

Author response: We thank the referee for his/her insightful questions and constructive suggestion. We totally agree with the referee that the cofactors act as the electron carriers to transfer electrons between a photocatalyst and enzyme, requiring the direct interaction between cofactors and Tk. If the TK is completely shielded by the decoration of CdTe, we assume that the electron collection and transport will be severely restricted. However, according to the TEM images of our prepared Tk–CdTe (the newly added Supplementary Fig. 7), the CdTe QDs are dispersed on TK membrane rather than complete coating. Thus, the direct interaction between cofactors and Tk will not be inhibited. This is also confirmed by the result that the performance of Tk–CdTe in cofactors regeneration exhibits a volcano profile when increasing the loading amount of CdTe QDs on thylakoid (Fig. 3a–c).

(18) The reaction mechanism and the nature of the active site are unclear. For example, what is the oxidation reaction?

Author response: We thank the referee for his/her thoughtful comment. According to the suggestion, considering that photosystem II (PSII) in thylakoid plays a key role in transferring protons and electrons, accompanied by cofactors generation in natural photosynthesis (Angew. Chem. Int. Ed., 2017, 56, 12903-12907), we have sought to unveil whether PSII underlies the accelerated electrons transfer in our Tk–CdTe module. To this end, PSII was isolated from thylakoid and combined with CdTe QDs (Supplementary Fig. 19 and Supplementary Table 5), denoted as PSII–CdTe, which showed superior electron transfer to Tk–CdTe. As shown in Fig. 4f, after combination with PSII, the lifetime of CdTe⁺ QDs shows a much faster decay (1.61 ns, Supplementary Table 6) compared to that in Tk–CdTe. Meanwhile, after combination with CdTe⁺ QDs, the lifetime of PSII increases from 0.86 ns to 1.48 ns (Fig. 4g and Supplementary Table 7). These results confirm the electron transfer from the excited CdTe QDs to PSII. Furthermore, the efficiency of DCPIP reduction by PSII–CdTe was superior to that by Tk–CdTe (Fig. 4h), which is most likely ascribed to the direct contact of CdTe QDs with PSII. As the conduction band (CB) of PSII is –0.58 V (vs. NHE) and the CB of CdTe QDs was measured to be –1.5 V (vs. NHE), the photoexcited electrons in CdTe QDs can be favorably injected into the CB of PSII through an interfacial charge-transfer process (Fig. 4i). The results collected above suggest that PSII could play an essential role in mediating electrons transfer between CdTe QDs and thylakoid. For CO₂ conversion reaction, the diverse reductases provide the active sites. Moreover, the reaction buffer contains sodium L-ascorbate as a hole acceptor, which can consume the photogenerated holes over CdTe QDs by oxidation reaction and prevent electron–hole recombination. The related discussion and data have been supplemented in the revised manuscript.

(19) The phrase "programmable" in the title cannot be supported by the reported context and is misleading. I strongly suggest the authors to modify the title.

Author response: We thank the referee for his/her kind suggestion. According to the suggestion, we have modified the title by removing the phrase "programmable" in the revised manuscript.

(20) The reported numbers for the photochemical CO₂ conversion should be comparable with other reports in the field. Can the authors estimate the number of consumed charge rate per active site for this biotic-abiotic hybrid (for instance, see Ref. [Nat. Commun., 13 (2022) 1256.]?)

Author response: We thank the referee for his/her thoughtful suggestion. According to the suggestion, we have compared the reaction rate of CO₂ conversion for this biotic–abiotic hybrid with the state-of-the-art studies on the cofactor-dependent photochemical CO₂ conversion (Supplementary Table 10), and our system showed a superior performance.

Supplementary Table 10 | The reaction rate for CO₂ conversion to C1 product in our system compared to the state-of-the-art studies on photochemical CO₂ conversion.

Reaction rate of CO₂ conversion to C1 product (μM mg⁻¹ h⁻¹)*	References
< 14.95	J. Am. Chem. Soc. 2014, 136, 16728-16731
0.54	Angew. Chem. Int. Ed. 2015, 54, 13971 –13974
1	J. Am. Chem. Soc. 2015, 137, 42, 13440–13443
0.22	Small 2016, 12, 4753-4762
96.35 ± 8.85	J. Am. Chem. Soc. 2018, 140, 16418–16422
82.29 ± 6.17	ACS Catal. 2019, 9, 3913-3925
< 18	J. Am. Chem. Soc. 2019, 141, 13, 5267–5274
177.25	ACS Appl. Mater. Interfaces 2020, 12, 34795–34805
< 9.78	Angew. Chem. Int. Ed. 2022, 61, e202200261
111.46	J. Am. Chem. Soc. 2022, 144, 31, 14207–14216
201.84 ± 23.89	This work

*Reaction rate was calculated as follows:

$$\text{Reaction rate} = \frac{\text{Concentration of C1 product}}{\text{Mass of catalyst} \times \text{Illumination time}}$$

Note: Mass of catalyst represents the sum mass of photocatalyst and CO₂ reductase.

As the cofactors are regenerated by photosynthetic protein in thylakoid membrane (e.g., ferredoxin-NADP⁺ oxidoreductase and ATP synthetase), the active sites are synergetic and complicated, which brings obstacles to estimate the number of active sites. Thus it is very difficult to estimate the number of consumed charge rate per active site for this biotic-abiotic hybrid. We sincerely hope that the referee could understand this situation.

Minor comments:

(1) As stated above, I do not see the motivation/rational of using CdTe QDs in this study. Do other QDs such as CdS, MoS₂, and others work in such a system? Besides, CdTe is known as toxic substance, why the author choose it as compared to other semiconductors?

Author response: We thank the referee for his/her thoughtful questions. In this work, as a proof of concept, we chose the CdTe QDs as a model to interface with thylakoid for constructing a novel biotic–abiotic hybrid energy module. We have also examined the potential of other well-studied colloidal semiconductor quantum dots, such as CdS and MoS₂ QDs, in constructing the hybrid energy modules for cofactors regeneration (the newly added Supplementary Fig. 20 and 21, Supplementary Table 8, 9). Both CdS and MoS₂ QDs exhibit good dispersibility, and the integrity of thylakoid structure can be well maintained after incorporation of CdS or MoS₂ QDs. Moreover, CdS and MoS₂ QDs also exhibit fluorescence quenching effects after combination with thylakoid. However, not only CdS and MoS₂ QDs but also their hybrid with thylakoid have a remarkably lower capacity for cofactors regeneration in contrast to CdTe QDs and its hybrid. Such an obvious distinction for cofactors regeneration among the different QDs suggest the unique superiorities of CdTe QDs for constructing hybrid energy module. In principle, compared with CdTe QDs, CdS and MoS₂ QDs own wide band gap and thus their light absorption is mainly in ultraviolet region (Supplementary Fig. 20 and 21), and as such, less electrons can be excited from the CdS and MoS₂ QDs under the simulated solar light.

(2) The authors should provide high-magnification TEM images displaying the presence of QDs on the thylakoids. The current TEM images (2nd column of Fig. 1a) are not clear. Actually, both this figure and that shown in Figure 2 look more like aggregated CdTe nanoparticles.

Author response: We thank the referee for his/her thoughtful suggestion. According to the suggestion, we have examined the presence of QDs on the thylakoids by high-magnification TEM images (Supplementary Fig. 7). The formation of Tk–CdTe hybrid structure through electrostatic interaction ensured homogeneous deposition of CdTe QDs on thylakoid, and negligible large CdTe⁺ aggregates were observed (Supplementary Figs. 2 and 7). Note that the observed bits of aggregates is due to the cross-linked PDADMAC rather than aggregation of CdTe⁺ (Supplementary Fig. 2a, b). No precipitate could be observed after centrifugation of the CdTe⁺ solution and it maintains a good dispersion, confirming that there is no aggregated CdTe nanoparticles (Supplementary Fig. 2c).

Supplementary Fig. 2 | Characterizations of CdTe⁺. (a) TEM image of CdTe⁺. The circles outlined by light dashed lines indicate bits of cross-linked PDADMAC and CdTe⁺. (b) HRTEM image of cross-linked PDADMAC and CdTe⁺. (c) Photograph of CdTe⁺ solution before and after centrifugation at 5,000 × g for 10 min.

Supplementary Fig. 7 | HRTEM image of CdTe⁺ located on the thylakoid. The circles outlined by red dashed lines indicate the magnified region.

(3) In the supplementary figure 1, What is the size of the QDs? Why the TEM images of the pristine CdTe QDs exhibit an aggregation after treatment with PDADMAC?

Author response: We thank the referee for his/her thoughtful questions. According to the suggestion, we have provided a higher-quality TEM image of CdTe QDs and plotted the histogram of the particle size in the revised manuscript (the updated Supplementary Fig. 1). Note that the observed bits of aggregates is due to the cross-linked PDADMAC rather than aggregation of CdTe⁺ (Supplementary Fig. 2a, b). No precipitate could be observed after centrifugation of the CdTe⁺ solution and it maintains a good dispersion, confirming that there is no aggregated CdTe nanoparticles (Supplementary Fig. 2c).

Supplementary Fig. 1 | Characterizations of positive charge CdTe QDs (CdTe⁺). (a) Schematic illustration for the preparation of CdTe QDs. (b) Photograph of prepared CdTe QDs dispersion. (c) TEM image of CdTe QDs. (d) Statistical size distribution of the as-synthesized CdTe QDs. (e) HRTEM image of CdTe QDs. (f) Zeta-potential profiles of CdTe QDs. (g) TEM image of CdTe⁺. (h) Statistical size distribution of CdTe⁺. (i) HRTEM image of CdTe⁺. (j) Zeta-potential profiles of CdTe⁺. The error bars represent the standard deviations of data from triplicate measurements.

(4) Did the authors perform any background blank tests for CO₂ photo-conversion experiments?

Author response: We thank the referee for his/her thoughtful question. In our original manuscript, we have carried out CO₂ photo-conversion experiments in artificial photosynthetic cells with several control groups including the background blank tests (Fig. 5f, Supplementary Figs. 51, 52 and Supplementary Table. 11). No product derived from CO₂ photo-conversion can be detected in background blank tests.

Reviewer #4:

In the manuscript “Programmable artificial photosynthetic cells with biotic-abiotic hybrid energy modules for customized CO₂ conversion” the authors report artificial photosynthetic cells where thylakoids are interfaced with CdTe quantum dots to obtain an enhanced flux of photogenerated electrons in the photosynthetic electron transfer cycle. This process is performed to drive an enhanced NADPH/NADH and ATP regeneration in the modified thylakoids. The regenerated cofactors are then used to drive CO₂ conversion, and a miniaturized system with artificial photosynthetic cells is also reported. The investigated topic, with the use of biohybrid systems for semi-artificial photosynthesis, is timely and of broad interest. Accordingly, the research topic is suitable for Nature Communications. However, there are various aspects in the manuscript that should be revised, as described in the specific comments below, and the work should be reconsidered after a major revision.

Author response: We really appreciate the referee’s highly positive evaluation of our work, and are grateful to the referee for his/her comments and suggestions to help us further improve the quality of our manuscript. We have made all the revisions as suggested by the referees.

Specific comments:

1) *Lines 171-200: When discussing the regeneration of NADPH, NADH and ATP by the thylakoid, CdTe QDs, and modified Tk-CdTe QDs, the comparison is performed in view of the results presented in Fig 3 a-c. However, Fig 3 d-f show that there is a clear influence of light intensity on cofactors regeneration. Accordingly, when discussing the results presented in Fig 3 a-c, it should be clearly mentioned that they refer only to a specific light intensity, since for low light intensities pristine Tk performed better than Tk-CdTe for NADPH regeneration (and similarly, CdTe had the same performance for NADH regeneration under low light intensity)*

Author response: We thank the referee for his/her thoughtful comment and suggestion. According to the suggestion, we have clearly mentioned the specific light intensity when discussing the regeneration of NADPH, NADH and ATP in the revised manuscript.

2) *lines 226-229: When discussing cofactors regeneration over multiple light/dark cycles, the decreased performance of CdTe QDs after 30 min is attributed to QDs aggregation, however, there is no experimental evidence supporting this claim.*

Author response: We thank the referee for his/her thoughtful comment. According to the comment, we have supplemented the experimental evidence of UV–Vis absorption spectra and TEM images (the newly added Supplementary Fig. 14, 15) to support the aggregation of individual CdTe QDs in the revised manuscript. As shown in Fig. 14a,

the UV–Vis absorption spectra of CdTe QDs has obvious change after 30 min light illumination. Moreover, the CdTe⁺ QDs display obvious aggregation in TEM image (Supplementary Fig. 15a) after 30 min light illumination. Meanwhile, the morphology and dispersibility of thylakoid and Tk-CdTe has no obvious change after 30 min light illumination.

Supplementary Fig. 14 | Photostability of Tk–CdTe after 30 min light illumination. (a–c) Normalized UV–Vis absorption spectra of (a) CdTe⁺, (b) Tk and (c) Tk–CdTe 1:1 in NADPH regeneration buffer before and after illumination. (d–f) Photographs of (d) CdTe⁺, (e) Tk and (f) Tk–CdTe 1:1 dispersion solution before and after illumination under the light intensity of 0.1 W/cm².

Supplementary Fig. 15 | TEM and SEM images of CdTe⁺, Tk and Tk–CdTe 1:1. (a) Before and (b) after 30 min illumination under the light intensity of 0.1 W/cm².

3) Lines 270-278: When discussing the bioactivity of the regenerated cofactors, it is stated that the non-enzymatic regeneration of the cofactors might result in inactive dimers or isomer. It is then stated that the cofactors regenerated by both pristine thylakoids and CdTe QDs were non-bioactive. While the results for CdTe QDs could be expected, it is not discussed why the enzymatically regenerated cofactors with pristine thylakoids are not bioactive.

Author response: We thank the referee for his/her thoughtful comment. Thylakoid is reasonable to produce bioactive NADPH. However, it requires a much higher amount of Tk (60 $\mu\text{g/mL}$ Chl equivalent) to produce detectable NADPH by ^1H NMR (Supplementary Fig. 26), which is remarkably inferior to Tk-CdTe. According to the comment, we have provided the related discussion and data in the revised manuscript.

Supplementary Fig. 26 | Bioactive assay of regenerated cofactors by Tk. The ^1H NMR spectra of standard (a) NADPH or (c) NADH, and the regenerated (b) NADPH or (d) NADH with corresponding NADP^+ or NAD^+ as a substrate in the presence of Tk at 60 $\mu\text{g/mL}$ Chl equivalent after 30 min illumination.

4) Lines 375-385: when presenting the cofactor regeneration performance in the artificial cells, it would be interesting to also compare NADPH regeneration, and not only NADH, since pristine Tk showed comparable performance to Tk-CdTe 1:1, especially under low light intensity.

Author response: We thank the referee for his/her thoughtful comment and suggestion. In the manuscript, we have presented the cofactors regeneration performance in the artificial cells, including NADH, NADPH and ATP (Fig. 5d, Supplementary Fig. 48, 49). Under light illumination (0.1 W/cm²), Tk–CdTe 1:1 demonstrated the superiority of cofactors regeneration in the artificial cells, compared to pristine Tk and CdTe QDs.

Fig. 5d | Light-driven NADH regeneration and CO₂ conversion in artificial photosynthetic cells. (d) NADH regeneration by CdTe⁺, Tk or Tk–CdTe with 80 μg/mL Chl equivalent in artificial photosynthetic cells after 10 min light illumination. Control denotes the counterpart only containing buffer E. The amounts of added CdTe⁺ was 69.12 μg/mL.

Supplementary Fig. 48 | NADPH regeneration by CdTe⁺, Tk or Tk–CdTe with 50 μg/mL Chl equivalent in artificial photosynthetic cells after 10 min light illumination. The amounts of added CdTe⁺ was 43.2 μg/mL. As a control, the counterpart only containing buffer F is included.

Supplementary Fig. 49 | ATP regeneration by CdTe⁺, Tk or Tk–CdTe with 50 μg/mL Chl equivalent in artificial photosynthetic cells after 10 min light illumination. The amounts of added CdTe⁺ was 43.2 μg/mL. As a control, the counterpart only containing buffer F is included.

Minor comments

- line 323 and 324: error values should be included for the percentages reported.

Author response: We thank the referee for his/her kind suggestion. In the revised manuscript, we have included the error values for the percentages reported as follow:

“...exceeding the production by Tk/PsFDH and CdTe⁺/PsFDH by 415.78 ± 87.72%

and $542.62 \pm 26.04\%$...”

- *in the introduction, the comparison of the approach presented in this work for enhancing the supply of photogenerated electrons to thylakoids with other approaches reported in literature is missing.*

Author response: We thank the referee for his/her thoughtful comment. We have accordingly checked the literature carefully. As we have stated in the introduction, the regeneration of cofactors is governed by energy-converting modules. Currently, two categories of energy modules—abiotic and biotic ones have been developed toward this purpose. The abiotic energy modules take advantage of the man-made materials (*e.g.*, semiconductors) to harvest light outpacing the natural photosynthesis. Nevertheless, most likely due to the deficient biocompatibility of synthetic materials, the electron transfer dynamics between light-harvesting energy modules and biological cofactors is severely hindered due to lack of electronic coupling. More seriously, the bioactive cofactors and enzymes tend to suffer from threat regarding to deactivation. Consequently, the abiotic energy modules commonly power NADH-dependent reactions (Supplementary Table 1) while lacking the capacity of regenerating multiple cofactors (*e.g.*, NADPH and ATP). In this work, we designed a programmable artificial photosynthetic cell by interfacing thylakoid with CdTe QDs as a novel biotic–abiotic hybrid energy module, which enabled efficiently regenerating multiple bioactive cofactors (NADH, NADPH and ATP) to power diverse photoenzymatic CO₂ conversion.

REVIEWER COMMENTS

Reviewer #1 (Remarks to the Author):

The authors made several revisions in this version. Generally, most questions and concerns were addressed. However, the key issue pertaining to the novelty and significance of this work still needs to be clarified. In line 209-214 and Fig 3b, it shows that the NADH regeneration can be observed when the Tk amount is 30-120 ug/mL Chl equivalent. In Fig S12, it can be found that there is a slight amount of NADH (less than 2 uM) with 10 ug/mL Chl Tk added. However, this can not be supported by their data in the supporting information. Based on the NMR spectra data in Fig S25d and FigS26d, when 15 or 60 ug/mL Chl Tk added, respectively, I cannot see any NADH. At least those very tiny peaks at 6.94 cannot convince me that there is NADH. They look like noises to me. It is widely known that natural Tk cannot regenerate NAD to NADH. Therefore, the data reported might be against this common sense. At least in my lab, we do not repeat this result. So please show us the original NMR data of the NADH regeneration, using 30-120 ug/mL Chl equivalent Tk, as shown in the Fig3b. It is highly possible that the Tk is not well prepared and other impurities may affect the result. Please rule out this possibility by doing some controls. If my guess is right, the rationale of this paper about the boosting effect of CdTe on the cofactor regeneration may not be solid. The novelty and the conclusion should be reconsidered.

Reviewer #2 (Remarks to the Author):

The authors have addressed most of my concerns, however, there are still several unanswered issues that need to be clarified for further consideration of publication in Nature Communications. My comments are listed below:

- (1) As shown in Supplementary Figure 23, the band gap of the CdTe is 1.86 eV. In addition, the authors noted that the light absorption of CdTe is not dominant compared to Tk (Supplementary Figure 8). Therefore, why do the authors expect charge transfer from the CdTe to the PS II component (e.g., shown in Figure 4i), plays a significant role in the photochemical CO₂ conversion process if the electron transfer or light absorption from or in CdTe is not dominant? What happens to the absorbed light in the PS II component? The presence of an interface between the CdTe and PS II may also play some role too. Anyway, the role of CdTe is still not clear or not addressed fully here yet.
- (2) What happens to the generated hole in the PS II component? Does it transfer to the CdTe? Where is the oxidation level of sodium L-ascorbate? I suggest the authors to address this process in the suggested schematic presented in Figure 4i. The authors are advised to illustrate the constituent components' band positions with respect to the reduction and oxidation levels of the CO₂ conversion.
- (3) Supplementary Table 10: the authors are suggested to add the corresponding catalyst system for each reference (as additional column). In my previous review report, I have asked the authors to calculate the estimated number of consumed charge rates per active site (R_e) for their biotic-abiotic hybrid; here in this revision, the authors provided the reaction rate for CO₂ conversion (which is not an intrinsic property due to the inherent difference in the actual active sites of the material system investigated), though I could understand the authors' explanation on the difficulty in estimating the number of consumed charge rates per active site. Alternatively, as a fundamental number assisting the readers to compare the results with other reported photocatalysts, perhaps the authors can provide the overall apparent quantum efficiency (AQE) or internal quantum efficiency (IQE). At least, the calculation of the AQE is quite straightforward. The authors may see Nat. Commun., 13 (2022) 1256 as a reference where R_e , AQE, and IQE are discussed in detail.
- (4) As explained in the first reference the authors cited in their response letter to Reviewer #2, Science 373, eaaz8541 (2021), the bandgap of quantum dots can be adjusted over a wide range of energies, spanning from ultraviolet to infrared wavelengths and the minimum value of the bandgap is determined by the bandgap of the parental bulk material from which the quantum dots are derived. Apart from the bandgap opening, the actual band positions (VBM and CBM), relative to those of its heterojunction (or hybrid), as well as the reduction and oxidation levels of CO₂ conversion will matter, as suggested in point 2 mentioned above. In the second reference, Science 352, 448-450 (2016), CdTe nanoparticles (actually, nanorod-like structure), not quantum dots, were used. Here, the authors should clarify the bandgaps and band positions for their specific purpose and provide more details on these physical parameters of their QDs with tunable size.
- (5) While I did not request additional experimental evidence or theoretical calculation, information

regarding the nature of the adsorption sites is indeed quite crucial for giving a rationale/guideline behind the design of the Abiotic-Biotic system. Such study can also help addressing the loading efficacy of CdTe+ in TK-CdTe.

(6) For question #7 in the review report (from Reviewer #2) on the volcano profile: Where exactly the authors included the related discussion in the revised manuscript?

(7) For question #13 in the review report (from Reviewer #2) on the energy diagram of the hybrid: Illustration of the oxidation process in the energy diagram (Fig. 4i) is still incomplete/confusing (see above-mentioned point 2). In the revised schematic in Fig. 4i, the authors seem to state that the conversion of NADP+/NAD+ to NADPH/NADH occurs on CdTe, and the conversion of ADP to ATP occurs on TK. Please clarify!

Reviewer #4 (Remarks to the Author):

During the revision process the authors have carefully addressed the comments of the reviewers and critically improved the manuscript. Specifically, new experimental evidence has been provided, including temperature monitoring of the reaction system, new UV-vis spectra, and high-magnification TEM, together with an improved discussion of the results.

The significant figures in the percentage values and errors must be corrected prior to publication (i.e., $415.78 \pm 87.72\%$ should be $420 \pm 90\%$).

I consider this work suitable for the community of Nat. Comm., and thus, after the revision of the significant figures, I have no restrictions to endorse the manuscript for publication.

Reviewer #1:

The authors made several revisions in this version. Generally, most questions and concerns were addressed. However, the key issue pertaining to the novelty and significance of this work still needs to be clarified. In line 209-214 and Fig 3b, it shows that the NADH regeneration can be observed when the Tk amount is 30-120 ug/mL Chl equivalent. In Fig S12, it can be found that there is a slight amount of NADH (less than 2 uM) with 10 ug/mL Chl Tk added. However, this cannot be supported by their data in the supporting information. Based on the NMR spectra data in Fig S25d and FigS26d, when 15 or 60 ug/mL Chl Tk added, respectively, I cannot see any NADH. At least those very tiny peaks at 6.94 cannot convince me that there is NADH. They look like noises to me. It is widely known that natural Tk cannot regenerate NAD to NADH. Therefore, the data reported might be against this common sense. At least in my lab, we do not repeat this result. So please show us the original NMR data of the NADH regeneration, using 30-120 ug/mL Chl equivalent Tk, as shown in the Fig3b. It is highly possible that the Tk is not well prepared and other impurities may affect the result. Please rule out this possibility by doing some controls. If my guess is right, the rationale of this paper about the boosting effect of CdTe on the cofactor regeneration may not be solid. The novelty and the conclusion should be reconsidered.

Author response: We are grateful to the referee for his/her insightful comments and suggestions to help us further improve the quality of our manuscript. According to the comments and suggestions, we have supplemented the control experiments, related data and discussion in the revised manuscript, and hope that we have addressed his/her concerns.

Due to the low reaction rate, using 5-120 $\mu\text{g/mL}$ Chl equivalent Tk can only regenerate a tiny amount ($< 20 \mu\text{M}$) of NADH, which is not detectable by the ^1H NMR spectra (this is confirmed by collect the ^1H NMR spectra of standard NADH solutions with different concentrations, Fig. R1). Thus, no clear characteristic signal was observed in ^1H NMR spectra for the NADH regenerated by pristine thylakoid. **Please note that the NADH regeneration activity was commonly assayed by spectrophotometry measurement** (please refer to the experimental details in the Methods of main text). In sharp contrast, the characteristic signal of NADH at 6.94 was unambiguously resolved for Tk–CdTe in ^1H NMR spectra. In our last round of revision, we have shown amplified NMR spectra in Supplementary Fig. 29d and 30d. The original intact NMR spectra of the NADH regeneration by using 60 $\mu\text{g/mL}$ Chl equivalent Tk is now uploaded and named “Original NMR data”.

Fig. R1. The ^1H NMR spectra of standard NADH solutions with different concentrations (0, 12.5, 25, 50, 100 and 200 μM).

It is widely known that natural Tk produces NADPH through photosynthetic electron transfer chains process (the updated Fig. 4i and newly added Supplementary Fig. 23). Specifically, ferredoxin-NADP⁺ reductase (FNR) (in Tk membrane) catalyzes the reversible exchange of reducing equivalents between the NADP⁺/NADPH and the ferredoxin (Fdx, Fe³⁺)/ferredoxin (Fdx, Fe²⁺) redox couples in photosynthetic reactions, which is the last step for generating NADPH (Arch. Biochem. Biophys. 2008, 474, 283–291). As FNR is usually highly specific for NADP⁺, it displays significantly higher reaction activity for NADPH regeneration in contrast to NADH regeneration (Arch. Biochem. Biophys. 2008, 474, 283–291; Arch Microbiol. 2004, 182: 80–89). It has reported that FNR can also catalyze the oxidized reaction of NADH/NAD⁺ with slower reaction rate compared with NADPH/NADP⁺ (Biochemistry. 2012, 51, 3819–3826). Moreover, NAD⁺/NADH shows the same redox potential as NADP⁺/NADPH (Bioelectrochemistry 2011, 80, 121–127; Front Microbiol. 2019; 10:866). Thus, most likely Tk can catalyze the reduction of NAD⁺ to NADH but with a low efficiency.

We further verified that the FNR in Tk catalyzes the reduction of NAD⁺ to NADH by evaluating the NADH regeneration capability in the presence of ferricyanide. It has reported that ferricyanide (as an electron acceptor) is photo-reduced primarily by FNR-contained photosystem I (PSI) (Plant Sci. Letters. 1976, 7, 171-178; Ann. Rev. Plant

Physiol. 1974. 25:423-58). Our experiment results show that the NADH regeneration capability by Tk is significantly reduced when the ferricyanide is introduced as a competitive electron acceptor (the newly added Supplementary Fig. 14), indicating that the reduction of NAD^+ to NADH occurs at the same FNR in PSI that produces NADPH. The above results suggest that regeneration NAD^+ to NADH with Tk is reasonable and factual.

Supplementary Fig. 23 | Cofactors regeneration and CO₂ reduction. (a) NAD(P)H and ATP regeneration via electron transfer chain and their redox potential levels to the CO₂ conversion. (b) Schematic illustration of a possible mechanism for electron transfer from excited CdTe⁺ to photosystem. Upon light illumination, the energy modules regenerate bioactive NADPH, NADH and ATP cofactors, which can be coupled with various cofactors-dependent CO₂ reductases to drive CO₂ conversion. SA: sodium L-ascorbate. Mn, Tyr, P680, Ph, Q_A, Q_B, PQ, Cyt, PC, and P700 are the abbreviation of the electron transfer factors in photosystem. Fdx: Ferredoxin. FNR: Ferredoxin-NADP⁺ oxidoreductase. NaDT: sodium dithionite. FDH: formate dehydrogenases. MoFe: remodeled nitrogenase.

Supplementary Fig. 14 | NADH regeneration by Tk with 0.1 mM K₃Fe(CN)₆ treatment.

Supplementary Fig. 5 | **Characterizations of thylakoid.** (a) BN-PAGE analysis of photosynthetic apparatus isolated from spinach. The thylakoid was solubilized by 2 % n-Dodecyl-β-D-Maltopyranoside (DDM). Thylakoid loaded in lane was 12.5 μg (equivalent of Chl). PSI: photosystem I; PSII: photosystem II; Cyt *b6f*: cytochrome *b6f* complex; LHCII: light-harvesting complex of photosystem II; CP43: photosystem II chlorophyll binding subunit. (b) UV-vis absorption spectra of Tk in 80% acetone. The Chl a/b ratio in our Tk was calculated to be 2.9.

The possibility that Tk is not well prepared and other impurities may affect the result has been ruled out. Tk was prepared by isolating from young spinach according to the

reported method (Science 2020, 368, 649-654; Angew. Chem. Int. Ed. 2022, 61, e202111054). Tk acquisition was performed by breaking the chloroplast in osmotic shock buffer, which obtained intact thylakoid and plasma membrane of broken chloroplast. Further purification was carried out by throwing away broken membrane with repeated washing. The light-absorbing antennas in Tk are chlorophyll (Chl) a and b. According to the method described by Porra (Photosyn. Res. 2002, 73, 149-156), the Chl a/b ratio in our thylakoid was calculated to be 2.9, which is consistent with the reported literatures (Angew. Chem. Int. Ed. 2022, 61, e202111054; Photosyn. Res. 2002, 73, 149-156) (the newly added Supplementary Fig. 5b). Taken together with BN-PAGE analysis (Supplementary Fig. 5a), the results suggest that the intact thylakoid with well-preserved structure and light-harvesting function was obtained in vitro.

Then, we have further performed some control experiments to eliminate the influence of possible impurities (e.g., plasma membrane, broken liposome and protein/proteoliposome) on NADH regeneration. The presence of plasma membrane in the supernatant of broken chloroplast buffer was verified by ferricyanide photoreduction assay (<https://www.sigmaaldrich.cn/deepweb/assets/sigmaaldrich/product/documents/638/446/cpisopis-mk.pdf>). As shown in the newly added Supplementary Fig. 15a, b, ferricyanide was partially photo-reduced due to the interference of intact plasma membrane without osmotic shock buffer treatment, confirming the presence of plasma membrane. We obtained the plasma membrane by collecting the supernatant in broken chloroplast with osmotic shock buffer treatment (the newly added Supplementary Fig. 15c) to estimate its influence on NADH regeneration. Similarly, we collected the broken liposome and protein/proteoliposome from thylakoid via n-Dodecyl- β -D-Maltopyranoside (DDM)-mediated method (Nat Biotechnol 2018, 36, 530–535; Nat Protoc. 2015, 10, 1328–1344) to estimate their influence on NADH regeneration (the newly added Supplementary Fig. 15c). The results demonstrate that the possible impurities have negligible NADH regeneration capacity and also have no significant influence on NADH regeneration by Tk (the newly added Supplementary Fig. 15d).

Supplementary Fig. 15 | The isolation of possible impurities and its influence on NADH regeneration by Tk. (a) UV-vis absorption spectra of $K_3Fe(CN)_6$ in Millipore water. (b) $K_3Fe(CN)_6$ photoreduction assay over Tk with/without osmotic shock buffer treatment. Note: the presence of intact plasma membrane in Tk without osmotic shock buffer treatment hinders the photoreduction of $K_3Fe(CN)_6$. (c) Possible impurities in the process of Tk isolation, containing broken plasma membrane, liposome and protein/proteoliposome. Supernatant: broken plasma membrane-contained buffer. Tk + DDM: isolated Tk solubilized by 2 % DDM. Liposome: precipitate via centrifugation from solubilized Tk. Tk + DDM + Bio-Beads: Bio-Beads added into solubilized Tk mixture. Protein/proteoliposome: reconstituted protein via removing DDM. (d) NADH regeneration by Tk with impurities (plasma membrane, liposome and protein/proteoliposome) treatment. The error bars represent the standard deviations of data from triplicate measurements.

Overall, the results collected above suggest that our Tk was well prepared with well-

preserved structure and light-harvesting function. The possibility that other impurities may affect the NADH regeneration has been ruled out. Although the factual result that Tk can catalyze the reduction of NAD^+ to NADH might be against the common sense, its extremely low efficiency will not influence the rationale, novelty and conclusion of our work about the boosting effect of CdTe QDs on the cofactor regeneration. The rational integration of thylakoid with CdTe QDs substantially promotes the regeneration of bioactive NADPH, NADH and ATP cofactors without requiring external supplements, remarkably exceeding the individual component (Fig. 3a-c). As such, we believe that our work is suitable for publication in Nature Communications.

Reviewer #2:

The authors have addressed most of my concerns, however, there are still several unanswered issues that need to be clarified for further consideration of publication in Nature Communications.

Author response: We are grateful to the referee for his/her insightful comments and suggestions to help us further improve the quality of our manuscript. We have made all the revisions as suggested by the referees.

My comments are listed below:

(1) As shown in Supplementary Figure 23, the band gap of the CdTe is 1.86 eV. In addition, the authors noted that the light absorption of CdTe is not dominant compared to Tk (Supplementary Figure 8). Therefore, why do the authors expect charge transfer from the CdTe to the PS II component (e.g., shown in Figure 4i), plays a significant role in the photochemical CO₂ conversion process if the electron transfer or light absorption from or in CdTe is not dominant? What happens to the absorbed light in the PS II component? The presence of an interface between the CdTe and PS II may also play some role too. Anyway, the role of CdTe is still not clear or not addressed fully here yet.

Author response: We thank the referee for his/her insightful comments and suggestions. The relative dominant light absorption of Tk is due to the antenna pigments that are not photochemically active. Tk photosynthesis is constrained to the visible range of the solar spectrum, allowing access to only roughly 50% of the incident solar energy radiation and less than 10% of full solar light saturates the capacity of the photosynthetic apparatus (Annu. Rev. Energy. 1979, 4, 353–401; Annu. Rev. Plant Biol. 2010, 61, 235–261). In fact, the intrinsic solar light absorption is ascribed to chlorophyll a and b, which exhibits weak capability to capture solar energy (the newly added Supplementary Fig. 5b). Thus, Tk generally suffers from low utilization of sunlight, which not only limits the efficiency of the whole photosynthetic process, but also poses a challenge for increasing the supply of photogenerated electrons (Nat. Nanotechnol. 2018, 13, 890-899). In this regard, we envisaged that efficient regeneration of cofactors can be achieved by rationally combining thylakoid and light harvesting inorganic nanomaterials to increase the supply of photogenerated electrons. Among various light-harvesting inorganic nanomaterials, the light absorption and electronic band structure of quantum dots are more readily tuned by size control. Quantum dots also have more negative conduction band potential compared to the corresponding nanoparticles, which could facilitate the photogenerated electrons transfer to thylakoid (Science 2021, 373, eaaz8541; Science 2016, 352, 448-450). Thus, we chose CdTe quantum dots for integration with Tk to increase the supply of photogenerated electrons. The charge transfer from the CdTe to the PSII component and its significant role in the photochemical CO₂ conversion process have been verified by our experimental results (Fig. 4).

b

Supplementary Fig. 5 | Characterizations of thylakoid. (b) UV-vis absorption spectra of Tk in 80% acetone. The Chl a/b ratio in our Tk was calculated to be 2.9.

The electrons transfer process in PSII component has been illustrated in the updated Fig. 4i (Nat Commun. 2014, 4 4647; Chem. Soc. Rev. 2014, 43, 6485-6497; Front. Microbiol. 2019, 10:866). Specifically, the photogenerated electrons from CdTe QDs transfer into the PSII through electron transport chain. Meanwhile, a photon inputs energy into the PSII by stimulating P680, which releases energetic electrons and produces the highly reactive P680* radical. The electron moves to pheophytin, and subsequently to electron transfer cofactors (Q_A and Q_B). P680 donates electrons and transforms into strong biological oxidizing agent (or electron acceptor), which captures the electron from a tyrosine residue (Tyr). The oxidized Tyr receives the electron from Mn via the splitting of water (H^+ and O_2 generation). This electron flow continues from photoexcited CdTe QDs and water oxidation to the PSII as long as Tk-CdTe is stimulated by photons.

We totally agree with the referee that the interface between CdTe QDs and PSII plays a key role in electron transport, which increases the supply of photogenerated electrons to boost the cofactors regeneration by Tk. The rational integration of thylakoid with CdTe QDs substantially enhances the regeneration of bioactive NADPH, NADH and ATP cofactors without external supplements by promoting proton-coupled electron transfer. According to the comments and suggestions, we have supplemented the related data and discussion in the revised manuscript.

Fig. 4 | (i) Schematic illustration of the electron transfer from CdTe⁺ to PSII with their corresponding redox potentials. Mn, Tyr, P680, Ph, Q_A and Q_B are the abbreviation of the electron transfer factors in PSII. SA is the abbreviation of sodium L-ascorbate.

(2) What happens to the generated hole in the PS II component? Does it transfer to the CdTe? Where is the oxidation level of sodium L-ascorbate? I suggest the authors to address this process in the suggested schematic presented in Figure 4i. The authors are advised to illustrate the constituent components' band positions with respect to the reduction and oxidation levels of the CO₂ conversion.

Author response: We thank the referee for his/her thoughtful comments and suggestions. As we have stated in above response, the electrons transfer process in PSII component has been illustrated in the updated Fig. 4i. The photogenerated hole by P680 is more likely finally consumed on Mn for oxidization of water (H⁺ and O₂ generation) in PSII, thus we believe that it will not transfer to the CdTe. The oxidation potential of sodium L-ascorbate is 0.2 V vs. NHE (Org. Biomol. Chem. 2017,15, 4417) which is suitable to recombine with the photogenerated hole of CdTe QDs (the updated Fig. 4i). According to the suggestion, we have addressed this process in the updated Fig. 4i, and have illustrated the constituent components' band positions with respect to the reduction and oxidation levels of the CO₂ conversion (the newly added Supplementary Fig. 23a).

Overall, CdTe QDs and PSII transfer photogenerated electrons to PSI via electron transport chain, and the photogenerated electrons ultimately are accepted by NAD(P)⁺ to produce NAD(P)H. During the electrons transfer process, a proton gradient, which is derived from the electron transport chain (i.e., oxidizing of reduced plastoquinone) or splitting of water, is generated across the thylakoid membrane, driving the synthesis of ATP (Jakubowski, H.; Flatt, P. Light Absorption in Photosynthesis-An Overview <https://bio.libretexts.org/@go/page/15046>). Finally, the produced cofactors of NADPH,

NADH and ATP are used to reduce CO₂ via CO₂ reductase (the updated Fig. 4i and the newly added Supplementary Fig. 23). NADPH or NADH as reducing equivalents are utilized by FDH to convert CO₂ into HCOO⁻. ATP is utilized by MoFe to power the conversion of CO₂ into CH₄ through combining with NaDT, which has an oxidation potential of -0.66 V vs. NHE (J. Am. Chem. Soc. 2021, 143, 18159-18171) and is typically used as an electron donor for MoFe protein catalysis (Proc. Natl. Acad. Sci. U.S.A. 2012, 109, 19644-19648; J. Am. Chem. Soc. 2023, 145, 5637-5644). According to the comments and suggestions, we have supplemented the related data and discussion in the revised manuscript.

Supplementary Fig. 23 | Cofactors regeneration and CO₂ reduction. (a) NAD(P)H and ATP regeneration via electron transfer chain and their redox potential levels to the CO₂ conversion. SA: sodium L-ascorbate. Mn, Tyr, P680, Ph, Q_A, Q_B, PQ, Cyt, PC and P700 are the abbreviation of the electron transfer factors in photosystem. Fdx: ferredoxin; FNR: ferredoxin-NADP⁺ oxidoreductase. NaDT: sodium dithionite. FDH: formate dehydrogenases. MoFe: remodeled nitrogenase.

(3) *Supplementary Table 10: the authors are suggested to add the corresponding catalyst system for each reference (as additional column). In my previous review report, I have asked the authors to calculate the estimated number of consumed charge rates per active site (R_e) for their biotic-abiotic hybrid; here in this revision, the authors provided the reaction rate for CO₂ conversion (which is not an intrinsic property due to the inherent difference in the actual active sites of the material system investigated), though I could understand the authors' explanation on the difficulty in estimating the number of consumed charge rates per active site. Alternatively, as a fundamental number assisting the readers to compare the results with other reported photocatalysts, perhaps the authors can provide the overall apparent quantum efficiency (AQE) or internal quantum efficiency (IQE). At least, the calculation of the AQE is quite*

straightforward. The authors may see *Nat. Commun.*, 13 (2022) 1256 as a reference where R_e , AQE, and IQE are discussed in detail.

Author response: We thank the referee for his/her valuable comments and suggestions. According to the suggestion, we have added the corresponding catalyst system for each reference (as additional column), and compared the quantum efficiency for this biotic-abiotic hybrid with the state-of-the-art studies on the photochemical CO₂ conversion in Supplementary Table 10. We have also cited the reference (ref. 58) provided by the referee in the revised manuscript.

Internal quantum efficiency was calculated as follows (please refer to the calculation details in Supplementary Information):

$$\text{IQE} = \frac{\text{the number of electrons for production of HCOOH}}{\text{total absorbed incident photons}}$$

Supplementary Table 10 | The reaction rate and quantum efficiency for CO₂ conversion to C1 product in our system compared to the state-of-the-art studies on photochemical CO₂ conversion.

Catalyst system	Reaction rate ($\mu\text{M mg}^{-1} \text{h}^{-1}$) ⁱ	Quantum efficiency ⁱⁱ	References
Treated rape pollen (TRP)	1.4	AQE 0.32%	Energy Environ. Sci. 2018, 11, 2382-2389
Photosystem II/ FDH tandem cell	96.35 ± 8.85	AQE < 0.033 ± 0.004%	J. Am. Chem. Soc. 2018, 140, 16418–16422
CdS-PTi/ TsFDH–FaldDH–YADH	82.29 ± 6.17	IQE 0.40 ± 0.08%	ACS Catal. 2019, 9, 5, 3913–3925
ZnPc/BVNS	< 0.01	AQE < 0.3%	Angew. Chem. Int. Ed. 2019, 58, 10873 –10878
CuIn ₅ S ₈ layers	N.P. ⁱⁱⁱ	AQE 0.786%	Nat. Energy. 2019, 4, 690-699
M _{0.33} WO ₃	< 18	N.P.	J. Am. Chem. Soc. 2019, 141, 13, 5267–5274
TPE-C ₃ N ₄ / FDH	177.25	AQE 0.227 ± 0.047%	ACS Appl. Mater. Interfaces 2020, 12, 34795–34805
^m CD/CN	0.099	IQE 2.1%	Nat Commun. 2020, 11, 2531
Reduced titania-Cu ₂ O	N.P.	AQE 0.012%	Appl. Catal. B. 2020, 279, 119344
3DOM CdSQD/NC	104.2	AQE 2.9%	Adv. Mater. 2021, 33, 2102690
COF/FDH	< 9.78	AQE < 0.013%	Angew. Chem. Int. Ed. 2022, 61, e202200261
15CN@M	< 0.1	AQE 1.7%	J. Am. Chem. Soc. 2022, 144, 9576–9585
GCE/ α -CD	111.46	AQE 0.2 ± 0.1%	J. Am. Chem. Soc. 2022, 144, 14207–14216
2H-WSe ₂	N.P.	IQE 0.23%	Nat Commun. 2022, 13, 1256
Tk-CdTe/FDH	201.84 ± 23.89	IQE 2.46 ± 0.19%	This work

ⁱReaction rate was calculated as follows:

$$\text{Reaction rate} = \frac{\text{Concentration of C1 product}}{\text{Mass of catalyst system} \times \text{Illumination time}}$$

Note: Mass of catalyst system represents the mass of photocatalyst or the sum mass of light-absorption materials and CO₂ reductase.

ⁱⁱAQE: apparent quantum efficiency; IQE: internal quantum efficiency.

ⁱⁱⁱN.P.: not provided.

(4) As explained in the first reference the authors cited in their response letter to Reviewer #2, Science 373, eaaz8541 (2021), the bandgap of quantum dots can be adjusted over a wide range of energies, spanning from ultraviolet to infrared wavelengths and the minimum value of the bandgap is determined by the bandgap of the parental bulk material from which the quantum dots are derived. Apart from the bandgap opening, the actual band positions (VBM and CBM), relative to those of its heterojunction (or hybrid), as well as the reduction and oxidation levels of CO₂ conversion will matter, as suggested in point 2 mentioned above. In the second reference, Science 352, 448-450 (2016), CdTe nanoparticles (actually, nanorod-like structure), not quantum dots, were used. Here, the authors should clarify the bandgaps and band positions for their specific purpose and provide more details on these physical parameters of their QDs with tunable size.

Author response: We thank the referee for his/her thoughtful comments and suggestions. As we have responded to Point 1 and 2, according to the suggestion, we have clarified the electrons transfer process in PSII component, and have illustrated the constituent components' band positions with respect to the reduction and oxidation levels of the CO₂ conversion (the updated Fig. 4i and the newly added Supplementary Fig. 23a).

As we have explained in previous response, among various light-harvesting inorganic nanomaterials, the light absorption and electronic band structure of quantum dots are more readily tuned by size control. Quantum dots have more negative conduction band potential compared to the corresponding nanoparticles, which could facilitate the photogenerated electrons transfer to thylakoid (Science 2021, 373, eaaz8541). Specifically, the more negative conduction band potential of CdTe QDs supplies more energetic photoexcited electrons and facilitates the photogenerated electrons transfer to PSII. Meanwhile, the appropriate valence band potential of CdTe QDs also facilitates the recombination of photogenerated holes with electrons from biocompatible sodium L-ascorbate to promote electrons separation. Thus, we chose CdTe QDs for integration with thylakoid.

Quantum confinement, leading to size-dependent optical and electrical properties that are distinct from those of the parental bulk solids, occurs when the spatial extent of electronic wave functions is smaller than the Bohr exciton diameter (a_B). As QDs become smaller, quantum confinement increases the effective bandgap. Specifically, the energies of the valence band maximum (VBM) will gradually decrease, while the

conduction band minimum (CBM) will gradually increase, leading to a blue shift of the absorption and emission spectra (Science 2021, 373, eaaz8541; J. Phys. Chem. C. 2016, 120, 650-658; Nano Energy. 2018, 46, 45-53) (the newly added Supplementary Fig. 1). According to the suggestion, we have included the related discussion in the revised manuscript.

Supplementary Fig. 1 | Illustration of the quantum confinement leading to size-dependent optical and electrical properties.

(5) While I did not request additional experimental evidence or theoretical calculation, information regarding the nature of the adsorption sites is indeed quite crucial for giving a rationale/guideline behind the design of the Abiotic-Biotic system. Such study can also help addressing the loading efficacy of CdTe⁺ in TK-CdTe.

Author response: We thank the referee for his/her helpful comments and suggestions. We totally agree with the referee that the adsorption sites are indeed quite crucial for designing of the Abiotic-Biotic system. The thylakoid membrane mainly contains lipids and photosynthetic protein complexes, and exhibits negative surface charge mainly attributed to the exposed charged amino acids of the photosynthetic protein (Cell Biochemistry and Biophysics, 2020, 78:401-414; Biochimica et Biophysica Acta, 2008, 1778, 2823-2833). The presence of negative charges on thylakoid membrane enables the adsorption of positively charged CdTe⁺ through electrostatic force, which is confirmed by SEM, EDS mapping, CLSM and TEM images (Fig. 2 and Supplementary Fig. 8). Moreover, the effective adsorption sites should be beneficial for photogenerated electrons transfer, while the pure lipid bilayer has weak capacity for transferring electrons (Chem. Soc. Rev. 2021, 50, 4833–4855). Therefore, the adsorption sites for CdTe⁺ on thylakoid membrane could be the negative charged carboxyl in lipids or photosynthetic protein, whereas the effective adsorption sites for CdTe⁺ on thylakoid membrane are probably the negative charged carboxyl in photosynthetic protein.

In our manuscript, we have demonstrated the promoted electrons transfer for PSII-CdTe (Fig. 4h). Considering that PSII is the initial site of electrons transfer, accepting photogenerated electrons from CdTe⁺ (the updated Fig. 4i), we thus reasonably propose that the negative charged carboxyl of protein in PSII (LHCII complex and PSII core complex, Supplementary Fig. 22a) is likely the effective adsorption sites for CdTe⁺. We totally agree with the referee that determining the particular adsorption sites can help addressing the loading efficacy of CdTe⁺ in TK-CdTe. However, currently it is difficult to determine exactly the particular protein sites for CdTe⁺ due to the complex structure of proteins in PSII, thus requiring further exploration in future. We sincerely hope that the referee could understand this situation. According to the comments and suggestions, we have supplemented the related discussion in the revised manuscript.

(6) For question #7 in the review report (from Reviewer #2) on the volcano profile: Where exactly the authors included the related discussion in the revised manuscript?

Author response: We thank the referee for his/her kind suggestion. In the previous revised manuscript, we included the related discussion for the NADPH regeneration that has the similar trend when increasing the loading amount of CdTe QDs. According to the suggestion, we have included the related discussion for NADH and ATP regeneration in the revised manuscript as follows:

“Notably, when increasing the loading amount of CdTe QDs, the performance of Tk-CdTe in both NADH and ATP regeneration exhibited a volcano profile. This feature is most likely due to the limited amount of electron transfer medium in thylakoid or the hindered light absorption by over decoration of CdTe QDs.”.

(7) For question #13 in the review report (from Reviewer #2) on the energy diagram of the hybrid: Illustration of the oxidation process in the energy diagram (Fig. 4i) is still incomplete/confusing (see above-mentioned point 2). In the revised schematic in Fig. 4i, the authors seem to state that the conversion of NADP⁺/NAD⁺ to NADPH/NADH occurs on CdTe, and the conversion of ADP to ATP occurs on TK. Please clarify!

Author response: We thank the referee for his/her kind suggestion. We have noticed that the original Fig. 4i may mislead the referee to arouse confusion about the location for conversion of NADP⁺/NAD⁺ to NADPH/NADH. It should be noted that the round yellow balls represent CdTe⁺ and the oval yellow balls represent FNR. Thus, the conversion of NADP⁺/NAD⁺ to NADPH/NADH occurs on FNR in PSI. According to the comment, we have modified the related figures (the updated Fig. 4i and the newly added Supplementary Fig. 23b) by marking the round yellow balls with “CdTe⁺” in the revised manuscript.

Supplementary Fig. 23 | Cofactors regeneration and CO₂ reduction. (b) Schematic illustration of a possible mechanism for electron transfer from excited CdTe⁺ to photosystem. Upon light illumination, the energy modules regenerate bioactive NADPH, NADH and ATP cofactors, which can be coupled with various cofactors-dependent CO₂ reductases to drive CO₂ conversion. SA: sodium L-ascorbate. Mn, Tyr, P680, Ph, Q_A, Q_B, PQ, Cyt, PC and P700 are the abbreviation of the electron transfer factors in photosystem. Fdx: Ferredoxin. FNR: Ferredoxin-NADP⁺ oxidoreductase.

Reviewer #4:

During the revision process the authors have carefully addressed the comments of the reviewers and critically improved the manuscript. Specifically, new experimental evidence has been provided, including temperature monitoring of the reaction system, new UV-vis spectra, and high-magnification TEM, together with an improved discussion of the results.

The significant figures in the percentage values and errors must be corrected prior to publication (i.e., $415.78 \pm 87.72\%$ should be $420 \pm 90\%$).

I consider this work suitable for the community of Nat. Comm., and thus, after the revision of the significant figures, I have no restrictions to endorse the manuscript for publication.

Author response: We really appreciate the referee's highly positive evaluation of our work, and are grateful to the referee for his/her comments and suggestions to help us further improve the quality of our manuscript. According to the suggestion, we have corrected the significant figures in the percentage values and errors in the revised manuscript.

REVIEWERS' COMMENTS

Reviewer #1 (Remarks to the Author):

The authors have addressed all the concerns and now the revised version can be published.

Reviewer #2 (Remarks to the Author):

The authors have addressed my previous comments, significantly elevating the manuscript's quality. Noteworthy additions include a thorough discussion of the energy diagram of the thylakoid–CdTe quantum dots hybrid and elucidation of possible charge transfer pathways. The calculated IQE of $2.46 \pm 0.19\%$ underscores the system's substantial potential for photocatalytic CO₂ reduction applications, adaptable across diverse material designs. Furthermore, the authors' explanations of the challenges in pinpointing exact adsorption sites within this intricate system demonstrate a pragmatic perspective. Considering these substantial improvements and the anticipated impact on future research, I wholeheartedly recommend publication in Nature Communications.